# Determining the Damage and Failure Behaviour of Textile Reinforced Composites under Combined In-Plane and Out-of-Plane Loading

**DOI:** 10.3390/ma13214772

**Published:** 2020-10-26

**Authors:** Christian Düreth, Daniel Weck, Robert Böhm, Mike Thieme, Maik Gude, Sebastian Henkel, Carl H. Wolf, Horst Biermann

**Affiliations:** 1Institute of Lightweight Engineering and Polymer Technology (ILK), Technische Universität Dresden, Holbeinstraße 3, 01067 Dresden, Germany; daniel.weck@tu-dresden.de (D.W.); mike.thieme@tu-dresden.de (M.T.); maik.gude@tu-dresden.de (M.G.); 2Faculty of Engineering, Leipzig University of Applied Sciences, Karl-Liebknecht-Straße 134, 04277 Leipzig, Germany; robert.boehm.1@htwk-leipzig.de; 3Institute of Materials Engineering (IWT), Technische Universität Bergakademie Freiberg, Gustav-Zeuner-Straße 5, 09599 Freiberg, Germany; henkel@ww.tu-freiberg.de (S.H.); carl.wolf@ww.tu-freiberg.de (C.H.W.); biermann@ww.tu-freiberg.de (H.B.)

**Keywords:** textile reinforced composites, damage of composites, failure of composites, in-plane load, out-of-plane load, biaxial load application, finite element analysis, representative unit cell, multi-scale methods

## Abstract

The absence of sufficient knowledge of the heterogeneous damage behaviour of textile reinforced composites, especially under combined in-plane and out-of-plane loadings, requires the development of multi-scale experimental and numerical methods. In the scope of this paper, three different types of plain weave fabrics with increasing areal weight were considered to characterise the influence of ondulation and nesting effects on the damage behaviour. Therefore an advanced new biaxial testing method has been elaborated to experimentally determine the fracture resistance at the combined biaxial loads. Methods in image processing of the acquired in-situ CT data and micrographs have been utilised to obtain profound knowledge of the textile geometry and the distribution of the fibre volume content of each type. Combining the derived data of the idealised geometry with a numerical multi-scale approach was sufficient to determine the fracture resistances of predefined uniaxial and biaxial load paths. Thereby, Cuntze’s three-dimensional failure mode concept was incorporated to predict damage and failure. The embedded element method was used to obtain a structured mesh of the complex textile geometries. The usage of statistical and visualisation methods contributed to a profound comprehension of the ondulation and nesting effects.

## 1. Introduction

Reducing carbon emissions is one of today’s most important tasks in engineering applications. By 2050 the aim of European Commission according to the Paris Agreement is to cut greenhouse gas emissions—in particular, the fraction of carbon dioxide by 80% below 1990 levels in the next few years [1]—hereby reducing the structural weight of trains, cars, airplanes, etc., which is one of the key factors in reaching that goal. Using optimised carbon fibre reinforced plastics (CFRP) can lead to a significant weight reduction, which can consequently decrease carbon emissions up to 20% [2]. The applications and research projects of CFRP cover the construction industry with, for example, replacing steel reinforcements by lignin-based carbon fibre reinforced thermosets [3] or the retrofitting of masonry structures with carbon fibre meshes [4,5], the transportation industry with, for example, hybrid fibre reinforced thermoplastic hollow drive-shafts [6], up to high-performance applications in the aviation and space industry. Especially in the these industries, where weight is the most influencing factor regarding economic and ecological improvements, advanced materials such as textile reinforced composites are already standard [7]. In particular, carbon fibre reinforced textile composites are deployed, which can benefit from their high specific strength as well as stiffness, design freedom, high production rates and damage-tolerant behaviour [8]. Generic applications—e.g., fan-blades of a jet-engine or structural parts of the fuselage of an airplane, such as stringers—are to be mentioned. Both examples of advanced composite structures are subjected to a similar three-dimensional stress states in the vicinity of the load introduction area [9]. The superposition of a fibre-parallel tensile stress with an out-of-plane compressive stress is highly critical with respect to failure in these areas.

Whereas the damage and fatigue behaviour of uni-directional (UD) CFRP material is very advanced [10,11], the distinctive and heterogeneous failure and damage behaviour of textile reinforced composites has to be investigated more thoroughly to guarantee a more reliable and damage tolerant design of advanced structural parts. Authors like Böhm [12] or Tsai-Hill [13] can describe the in-plane failure and damage behaviour of textile reinforced composites by an invariant based or an interactive criteria sufficiently, respectively. An expedient description of the rate dependent visco-elastic and visco-plastic damage behaviour of textile-based glass fibre polypropylene is given in [14] implementing a rheological model and Böhm’s damage model. Recent advantages have been made in consideration of the out-of-plane compressive stress and the effect on the delamination behaviour of composites by Daniel [15] or Sun [16]. Nevertheless the specific textile architecture is neglected and all models describe the composite as a homogenised material on a macro-scopic scale. However, the influence of the textile architecture on the stiffness and strength behaviour have been analysed and determined in [17,18,19,20,21]. The commonly used methodology is a numerical multi-scale approach, which can benefit from advancements in numerical and experimental methods.

In this paper, a numerical and experimental multi-scale approach is presented. A new advanced testing method presented in [9,22] provides the opportunity to determine the fracture resistance of textile reinforced composites at predefined load combination of in-plane tensile and out-of-plane compressive stress. Most other testing methods for biaxial load application are restricted to prescribed load paths by geometry, have demanding requirements on the specimen design and preparation or cannot depict the size and the influence of the textile specific ondulation [23,24]. Hence, with this new testing method, profound data of pre-selected types of textile reinforcements at various load paths of the desired load application could be obtained [22]. For the determination of the influence of the inner structure, such as ondulation and nesting three plain-weave, fabrics with increasing areal weight have been considered. For comparability, equal weft and fill yarns have been chosen It was found that by increasing areal weight, the ondulation increases, thus so does the nesting effects change [25]. Based on the analysis of the compaction behaviour of the dry textile layers by in situ computer tomography [26,27,28] and image analysis of micrographs, profound knowledge of the geometry and arrangement of the textile reinforcements on meso-scale could be obtained. Additionally, the distribution of the yarns’ fibre volume content (FVC) on micro-scale could be determined by methods and algorithms in image processing and computer vision [29]. The FVC is an essential parameter to describe and to approximate the elastic properties and strength of composite material in analytical or numerical approaches [30,31].

This experimental knowledge of the fracture resistance, geometrical parameter and FVC will be used to analyse the damage and failure behaviour of three-dimensional loaded composites by a numerical multi-scale method. Convenient and commonly used methods to model the idealised geometry of a representative unit cell (rUC) of textile reinforcements are implemented in the two open-source applications WiseTex [32] and TexGen [33]. Both are integrated into the finite element software and solver Abaqus^®^, which has been used for the analysis. The state of art for the numerical description of the mechanical behaviour is to assume the yarns as a transversely isotropic unidirectional material with its specific FVC and orientation [34]. In the case of mesh quality and refinement, the method of the embedded elements (embedded element method; EEM) provided advantages in the numerical computation [35]. Regarding the load application of the periodic boundary condition of the rUC many works in literature are restricted to uniaxial tension or compression loading, regarding biaxial load applications most are in-plane stresses [36,37]. This paper will present and analyse the damage and failure behaviour of composites with selected textile reinforcements under combined load of in-plane tensile and out-of-plane compressive stress. In [17,36], progressive damage models are incorporated into the failure analysis. Nevertheless, those cannot provide differentiated information, a non binary (failure/no failure), about the damage mechanism. Therefore Cuntze’s invariant based failure criteria have been incorporated for the transversely isotropic assumed yarns. This criteria presented in [38] and revised in [39] are well known as the failure mode concept. It distinguishes between the five typical phenomenological failure modes (FM) of an UD-layer. In the world-wide failure exercise described in [40,41], the failure mode concept (FMC) achieved sufficient results. The experimental validation is presented by Cuntze in [42]. Additional, advanced evaluation methods, employed in the present work, for identifying the statistically distributed damage mechanism in discrete separated sections provide profound knowledge about the influence of the textile architecture at meso scale. Conclusively, an approach to the correlation or causal relationship between the obtained numerical and experimental data on different scales is provided to establish the basis for a reliable damage-tolerant design of textile reinforced composites exposed to combined in-plane tensile and out-of-plane compressive stresses.

The subsequent paper is distinguished in four Sections and auxiliary information is provided in Appendix A for Section 3 and in Appendix B for Section 4. In Section 2, the notation and the approach in numeric and experiment of each scale level is explained. Section 3 presents the experimental methods and Section 4 guides through the numerical methods. The comparison of the numerical and experimental determined fracture resistances is presented in Section 4.3.2. All results and findings are comprehensively explained and listed in Section 5.

## 2. Experimental and Numerical Multi-Scale Approach

The described multi-scale approach considers the three different scale levels—macro, meso and micro. Figure 1 illustrates the scale levels and notates each used Cartesian coordinate system (CS). The macro-scale uses the xyz- and wf3-CS to describe the behaviour and the results of the newly developed biaxial testing method ($\left(\sigma_{w}^{t}/\sigma_{3}^{c}\right)$-testing; Section 3.1) and the used standard testing methods (Section 3.3). To determine the influence of the ondulation and nesting behaviour of the considered textile reinforcements, considerations on the meso-scale are inevitable. The meso-scale level depicts the size of the weave pattern and is notated globally with the wf3-CS. The locally distributed 123-CSs refer to the modelling approach of the weft and fill yarns. Each yarn is depicted as a curved unidirectional layer (UD-layer). The behaviour of a UD-layer is sufficiently described by a fibre-matrix unit cell on the micro-scale. Thereby, the transversely isotropic behaviour is governed by the ‖⊥-CS and is invariant to a rotation about the ‖-axis.

At each scale, the paper describes experimental (Section 3) and numerical (Section 4) methods and their determined data or parameters. Commencing with the analysis of micrographs in Section 3.3.2, profound data of the distribution of the FVC ϕ and the fibre diameter $d_{f}$ could be obtained to predict the homogenised (〈〉 denotes the homogenised properties of an rUC) stiffness matrix $\langle C_{ijkl}\rangle$ and strengths $\langle R_{ij}^{c,t}\rangle$ of a fibre-matrix rUC (Section 4.1.3 and Section 4.2). These results are employed on the next higher scale (meso) to describe the behaviour of each weft and fill yarn, modelled as an individual curved UD-layer (Section 4.1.1). The idealised geometries and mean parameters, respectively, were derived from profound image analysis and processing of in situ CT measurements (Section 3.3.1; $\bar{\lambda },{\bar{t}}_{ply},{\bar{\beta }}_{c}$, etc.). The homogenised fracture resistances $\langle \sigma_{ij}\rangle$ of each rUC at combined in-plane and out-of-plane load were determined by Cuntze’s failure mode concept, described in Section 4.2, and compared to the experimental results of the new advanced biaxial testing method ($\left(\sigma_{w}^{t}/\sigma_{3}^{c}\right)$-testing; Section 3.1). After thorough validation of the numerical approach, an extensive statistical approach is presented in Section 4.3.3 and Section 4.3.4 to determine the influence of ondulation and nesting on the damage and failure behaviour of textile reinforced thermoset composites.

## 3. Experimental Methods

### 3.1. Biaxial Test Method for Textile Reinforced Composites: $\left(\sigma_{w}^{t}/\sigma_{3}^{c}\right)$-Testing

A novel and advanced specimen design and test set-up for biaxial load application of textile reinforced composites was developed [9]. The specimen design of the so called $\sigma_{w}^{t}/\sigma_{3}^{c}$-testing was adopted to a planar biaxial testing machine Instron 8800. The four orthogonally aligned servo-hydraulic actuators with a maximum load capacity of 250 kN per each were suitable for the desired stress state (Figure 2). Figure 3a illustrates the schematic test set-up of the ($\sigma_{w}^{t}/\sigma_{3}^{c}$)-testing. It provides that a tensile force *F* in *x*-direction evokes a fibre parallel in-plane tension stress $\sigma_{w}^{t}$ in the gauge section’s reference cross section ($A_{x,0}=tb$). The out-of-plane compressive stress $\sigma_{3}^{c}$ is introduced in the gauge section of the textile reinforced composite by numerically optimized compression stamps loaded with the compression force *P* (cf. Figure 3a, $A_{y,0}=lb$). A similar test set-up described in [24] with a cylindrical intender is characterised by a high stress concentration. Analytical solutions have been found in [43] to approximate the stress state.

Nevertheless, to sufficiently determine the influence of the specific textile architecture, the stress should be evenly distributed over the entire size of the specific textile representative unit. Therefore, each stamp is designed with a flat surface, the length *l* and rounded corners with radii *R* (Figure 3a). This ensures a homogeneous strain and stress distribution in the gauge section an,d due to the choice of large radii, it reduces the effect of stress peaks in the vicinity of the radii due to Hertzian contact [44]. For the experimental conduction, stamps with radius R=10 mm were machined from heat-treated maraging steel (Marage 300). This steel is characterised by a very high yield strength of 1815 MPa to withstand the high local contact pressure and a lower Young’s modulus of 193 GPa. In comparison to an estimated out-of-plane Young’s Modulus $E_{3}$ of ≈10 GPa, deformation of the stamp can be neglected. In order to reduce the friction and wear behaviour it was found that a diamond-like carbon coating was suitable [45]. Additionally an appropriate laminate thickness *t* could be determined in [46] to establish a homogeneous stress distribution and to reduce the influence of edge effects. Therefore a laminate thickness of t=10 mm was chosen. For consistency, the width *b* and length of the stamps’ flat surface *l* were set to the same size of 10 mm. All dimensions of the $\sigma_{w}^{t}/\sigma_{3}^{c}$-testing are illustrated in Figure 3b.

Due to the limited space of the testing machine and the high forces in specimen x-direction that can be encountered (with a $A_{x,0}=100$ mm and an expected in-plane tensile strength $R_{w}^{t}\approx 1000$ MPa, a maximum force of F≈100 kN is determined), a positive-locking wedge clamping has prevailed over conventional force-locking clamping (cf. Figure 2). To realise this wedge in the specimen design, ply-drops were employed in the manufacturing design. Sufficient design guidelines could be found in [47,48] to reduce stress concentrations and resin pockets. A non optimized ply drop-off design could therefore lead to a premature failure. Figure 3c shows the final $\sigma_{w}^{t}/\sigma_{3}^{c}$-specimen with magnification of the gauge section and the wedge clamping, including the ply drop off design.

To determine the stress–strain behaviour of the specimen, the stereo camera system Aramis^®^ 5M (*Gesellschaft für optische Messtechnik* (GOM), Braunschweig, Germany) was integrated into the test set-up and the test method ($\sigma_{w}^{t}/\sigma_{3}^{c}$-testing). The high resolution of the camera system enabled the opportunity to examine the deformation behaviour of the test set-up (Figure 4). This was necessary to compensate rigid body motion in the strain calculation by digital image correlation (DIC) of the specimen due to non-synchronous movement of the stamps. Therefore the region “fix area” near the constrained wedge clamping fixture was selected (Figure 4a) [49]. Furthermore, with the stereo camera system the displacement, $|{\bar{u}}_{y}|$ of the stamps was investigated more precisely. This was necessary to determine the increase in the stamps’ effective compression area through the stamp indentation. Figure 4b illustrates the $\varepsilon_{y}$ strain field with magnification on the vicinity of the stamp’s radius and its indentation in the specimen. This effect has to be taken into account for the stress calculations for both axes ($\sigma_{w}^{t}/\sigma_{3}^{c}$). A sufficient analytical method could be found to determine the true stress state and was validated through a numerical comparison [22]. A brief description of this method is provided in Section A.1.

### 3.2. Materials and Specimen Preparation

As mentioned, three types of plain-weave fabrics are used in this paper to distinguish the effects due to ondulation and nesting behaviour in the experimental and numerical analysis. The used textile reinforcements are listed in Table 1. All three ECC^™^-style fabrics are woven from a Tenax^®^ HTA 40 3K (200 tex, Heinsberg, Germany) yarn, but in a different setting. Consequently the weight of each specific textile reinforcement will increase, thus the ondulation will increase and the nesting behaviour will change [25,26].

The yarn consists of 3000 (3k) filaments, which are characterised by a high tenacity (HT). This reinforcement is commonly used in aviation applications. Table 2 lists selected properties of the filament.

The sufficiently analysed RTM6-2 from Hexcel^®^ was chosen as the matrix system of the textile-reinforced composites. This mono-component epoxy system is standard in aerospace industry applications because of its low processing viscosity and long gel time it is predestined for high quality parts with low void content. To complete the data, the matrix system RTM6-2 has a cured density $\varrho_{m}$ of 1.140 g cm^−3^ at 25 °C and a glass transition temperature $T_{g}$ of 190 °C.

The complex wedge shape as illustrated in Figure 3c could be realised in an in-house developed infiltration tool. Thereby it was possible to rely on profound knowledge in the thermal and infiltration design of resin transfer moulding (RTM) tools for the manufacturing of thick laminates [52]. Each laminate lay-up was set up to a macroscopic average fibre volume content (FVC) of $\bar{\phi }\approx$ 60%. From each infiltrated plate, 20 specimens, according to Figure 3b, were obtained by a subsequent abrasive cutting process. This guaranteed a high quality of the cutting edge with a smooth surface and the prevention from preliminary damage to the specimen. For quality assurance, all plates have been analysed from different regions regarding their FVC $\bar{\phi }$ and glass transition temperature ${\bar{T}}_{g}$.

### 3.3. Experimental Results of Uniaxial and Biaxial Testing

To determine the basic elasticity properties of the orthotropic textile reinforcements, uniaxial tests were used. The materials are assumed as symmetric in the wf-plane, according to Figure 1. The in-plane Young’s modulus $E_{w}$ and $E_{f}$, the in-plane major Poisson ratio $\nu_{wf}$ and strength $R_{w}^{t}$ and $R_{f}^{t}$, were obtained using a standardised test method according to DIN EN ISO 527-4 ($\sigma_{w}^{t}$-testing). A similar testing standard is the ASTM D3039. A modified test rig was used to conduct in-plane ($G_{wf}$) and out-of-plane shear ($G_{w3}$ and $G_{f3}$, respectively) properties according to ASTM D7078 (V-notched rail shear, VNRS) [53]. As mentioned before, for determining the out-of-plane strength ($R_{3}^{c}$) and elasticity properties ($E_{3}$, $\nu_{w3}$, and $\nu_{f3}$, respectively) of textile-reinforced composites, an advanced testing method described in [54] was used ($\sigma_{3}^{c}$-testing).

The results illustrated in Figure 5 and listed in Table A1 indicated a good agreement with the expected trend from analytical formulations in [55]. Thus, by increasing ondulation, the in-plane elastic properties will decline. The reverse trend in elastic out-of-plane properties could not be determined with sufficient accuracy. It could be assumed that, with higher ondulation, there is a higher fraction of fibres aligned in the direction of load and thus increase the properties. However, all statistical mean values lie within their confidence interval of one ±σ. Hence, there is no empirical or statistical evidence to sustain this hypothesis.

According to the uniaxial strengths determined for the load paths 1:0 and 0:1 ($\left\{{\tilde{\sigma }}_{L}\right\}$, Equation (Equation 2)) there is similar empirical evidence (cf. values for both load paths in Table A2). For load path 1:0, there is a decline of ≈2% by the maximal mean value,s as assumed according to higher ondulation in the in-plane tensile strength $R_{w}^{t}$ (corresponds to ${\bar{\sigma }}_{w,\max}^{t}$) for textile reinforcements I and II. For both types, the typical failure mode of fibre failure occurred (cf. Figure 6b). Type III showed a significantly other failure behaviour. Delamination, as shown in Figure 6c, started from the free edges of the specimen until fibre failure led to the final rupture. Hence, the values are not comparable. However, they emphasise the influence of the textile architecture on the damage and failure behaviour [18].

The uniaxial determination of the out-of-plane strength $R_{3}^{c}$ was in a good agreement with the observed behaviour in [46]. All types and specimens failed by the typical pyramidal failure mode (Figure 6a). This in good accordance to the observed failure pattern of the out-of-plane strengths testing methods presented in [15,42,46]. The deviation from the action and fracture plain could be analytically described by Puck in [57] for UD-material. As with Daniel, Puck describes the failure mechanism for out-of-plane compression as shear dominated, which causes the inclined failure pattern [15,50,57]. Cuntze incorporated this effect in the inter-fibre failure mode for transverse compression of UD-material. All explanations of this effect are based on a Mohr-Coulomb approach [50]. By comparison of the mean values ${\bar{R}}_{3}^{c}$ (Table A2) there is also no statistical evidence to conclude that a higher ondulation increases the out-of-plane properties. On the contrary, the newly developed and advanced $\sigma_{w}^{t}/\sigma_{3}^{c}$-testing was able to sufficiently determine the correlation between ondulation and fracture resistance. Figure 7a–c illustrate the testing results of the biaxial testing. All results have been determined by Equations (Equation 26) and (A2), which incorporate the stamp indentation, for the load paths 1:1, 1:2 and 1:5 according to Equation (Equation 2). All tests were performed in force control, hence, for the biaxial testing the denoted stress ratio vector $\left\{{\tilde{\sigma }}_{L}\right\}$ corresponds to the reference state ($A_{x,0}$ and $A_{y,0}$, cf. Equation (Equation 26) and (A2)) of the $\sigma_{w}^{t}:\sigma_{3}^{c}$-specimen. So that the stress vector reads to
(1)$$\begin{array}{cc} \left\{\sigma_{L}\right\} & ={\left[\sigma_{w,0}^{t},\sigma_{3,0}^{c}\right]}^{⊤}={\left[\frac{F}{A_{x,0}},\frac{P}{A_{y,0}}\right]}^{⊤} \end{array}$$
and the stress vector ratio for each load path to
(2)$$\begin{array}{cc} \left\{{\tilde{\sigma }}_{L}\right\} & =\sigma_{w,0}^{t}\;:\;\sigma_{3,0}^{c}=\frac{F}{A_{x,0}}\;:\;\frac{P}{A_{y,0}} \end{array}$$

Comparing the mean values of each load path in Figure 7d it can be stated that there is an influence of the textile architecture. Especially, load path 1:1 was characterised by an increase in the fracture resistance of 15% from type II to III. In comparison to type I with the lowest ondulation, an increase of 6% is still observed. There is no significant evidence that type II declines in the fracture resistance regarding type I. The assumed order of the corresponding fracture resistances (type I < II < III [55,58]) can be determined in the load paths 1:2 and 1:5. However, the increase by the mean values turns out to be less. At load path 1:2, there is a 10% and at load path 1:5 still a 6% increase in the fracture resistance (comparison of type I to III).

Figure 7d additionally classifies the corresponding failure modes to each specific load path. The uniaxial failure modes are listed for consistency. It could be observed for load path 1:1 and 1:2 that there is a pure fibre failure (cf. Figure 8a,b). The failure always occurred on the side of the load introduction (cf. Figure 3a). This relies on the frictional characteristic of the test set-up. The friction force between the stamp and the specimen reduces the axial force F. It could be determined that the reduction is governed by a linear Coulomb friction law with a friction coefficient μ. In comparison to the friction and wear properties of this material pairing from [45], it showed a reliable agreement. For load paths exceeding a load ratio of 1:2, a pyramidal inclined fracture pattern could be observed (cf. Figure 8c, load path 1:5).

#### 3.3.1. Meso-Scale Analysis of Textile Architecture

Sufficient numerical modelling depends on the reliability of obtained geometrical parameters [17,32,33]. Especially textile reinforced composites in particular fabric, reinforced composites reveal a high inherent distribution of their geometric features—e.g., ondulation. Furthermore the textile architecture particularly ondulation affects the arrangement and vice versa [25,59]. Hence the data of the advanced analysis method presented in [26] were suitable to derive essential geometric parameters. The compaction behaviour of dry textile reinforcements, according to Table 1, have been sufficiently analysed by in-situ computer tomography. Nevertheless, the spatial resolution of 20 µm per voxel of the acquired data by the FCTS 160-IS computer tomography did not provide the possibility to perform sufficient semantic segmentation to analyse the yarn course and parameters derived from it. Hence, an advanced image processing method of micrographs of consolidated specimens with $\bar{\phi }\approx$ 60% was elaborated to determine and to support the profound knowledge of all three textile reinforcements.

Figure 9a exemplarily illustrates the fitting results of one weft and the notation of the fittings. By a multi-level threshold analysis, the different regions, such as weft, fill and matrix, could be sufficiently semantically segmented by their grey value. This algorithm incorporates the heuristic Otsu-method for binarisation [60]. Subsequently, by an edge detection, the wefts’ outer contours could be highlighted. Each contour was approximated by a Fourier series of order one. In this way each yarn could be assigned a middle course by subtracting the outer counter fits. All the following necessary parameters, such as shifting between layers, ondulation, layer thickness etc., were derived from these processed data. For clarity, only two obtained parameters are exemplarily illustrated in the histograms of the probability density function in Figure 9a,b. The first parameter is the crimp angle $\beta_{c}$. It is defined as the pitch angle at the inflection point of the yarns’ course (Figure 9c) [59]. All types can sufficiently be approximated by a normal distribution ($f_{N}\left(x\right)$). The results of the distribution fits, the expectancy value $\mu_{N}$, the variance $\sigma_{N}^{2}$ and the coefficient of variation (CoV: $\nu =\frac{\sigma }{\mu }$) ν, are listed in Table 3. The second parameter is nesting factor by Potluri et al. ($\eta_{P}$). It is defined as
(3)$$\begin{array}{cc} \eta_{P} & =\frac{a_{0}}{\sum_{i=1}^{n}a_{i}}\;\forall \;n>2 \end{array}$$
and can assume values to a maximum value of 1. Thereby $a_{0}$ is the total thickness of *n*-layer and $a_{i}$ is the thickness of each individual layer [25]. That means that high value near 1 represents no nesting between layers. The selected types of textile reinforcements illustrate that a lower weight—consequently ondulation and crimp angle—leads to a higher nesting factor, which means less nesting capability (cf. Figure 9a,b). The fitting results of the nesting factor $\eta_{P}$$\mu_{N}$, $\sigma_{N}^{2}$ and the coefficient of variation are listed in Table 3.

#### 3.3.2. Microscopic Analysis of the Fibre Volume Content

To numerically describe the elastic properties with sufficient accuracy and approximate strengths the average microscopic FVC ${\bar{\phi }}^{\left(m\right)}$ of each yarn has to be determined [17]. Conventional methods that measure the FVC by removing the matrix by digestion or ignition according to ASTM D3171 or DIN EN 2564 provide evidence of a discrete volume at macro level. In that value, matrix-rich areas accumulate in the measured mean value and do not represent the local FVC in the yarn cross-section. Modern advanced methods in the area of image processing and computer vision of microscopic images are more expedient to sufficiently determine the local FVC. Commonly used algorithms are based on the evaluation of the grey value histogram [61]. Nevertheless these methods cannot determine the distribution of the average microscopic FVC ${\bar{\phi }}^{\left(m\right)}$ in the yarns’ cross-section. Hence, a more advanced approach has been developed. Figure 10 illustrates the schematic procedure.

The acquired microscopic images in grey scale are binarised by a heuristic threshold value determination. This so-called Otsu-method, introduced in [60], separates the matrix and fibres sufficiently to approximate the fibres’ circumference by the Houg circle transformation [29]. After extraction of each determined circle’s centroid, a Delaunay-triangulation was implemented to divide the area to be analysed into small sections—triangles (Figure 11a). According to [50], this represents the hexagonal packing, which hypothetically cannot exceed the value of $\phi =\pi 2\sqrt{3}\approx$ 91%.

Necessary assumptions are a perfect circle shape and equivalent radii. Nevertheless it could be experimentally determined that the fibres’ radii are subject to a normally distributed size with a small standard deviation ((6.85 ± 0.38) µm, fit over all used textile types). Finally, the local FVC $\phi^{\left(m\right)}$ could be calculated based on the area ratio of the circle segments to the corresponding total area of the triangle. The proposed algorithm was elaborated in the proprietary Software MatLab^®^, which was a convenient and fast method by using the ImageProcessing toolbox.

For this paper, micrographs of the three fabric types, each with three different positions, were prepared using a high-precision grinding and polishing process. All specimens have a global FVC of ≈60%. To obtain a high magnification (50×) and consequently high resolution, many overlapping images of a yarn were acquired to subsequently stitch them to a panorama. Thereby the proposed algorithm of [62] was used. The processed image, in Figure 11a had a final size of 39 to 47 Mpx.

The results of the analysis are given in Figure 11b,c as a histogram with overlayed probability density functions (f(x)) and as cumulative density function with overlayed distributions (F(x)) for all textile reinforcement types. For the evaluation of this randomly distributed variable ϕ, the Weibull (W) and normal distribution (N) was chosen. Both distributions approximate the randomly distributed variable sufficiently. They both reject the null hypothesis at significance level of 5% of the $\chi^{2}$-test. However, the Weibull distribution gains a higher value of the test statistic, which corresponds to a likely better approximation of ϕ.

Table 4 lists all results of the evaluation. Thereby, the mean (μ) and variances ($\sigma^{2}$) of the Weibull (W)- and normal (N)-distribution for each textile reinforcement and for an approximation of all data. It turns out that the deviation between the reinforcement is negligible and consequently the expectancy values μ=0.726 will be used for the stiffness and strength approximation in Section 4.

## 4. Numerical Methods

### 4.1. Numerical Modelling

#### 4.1.1. Geometry

The geometrical modelling of textile reinforcements is essential for the determination of realistic parameters by numerical methods [32,33,63]. As the experimental determination of the meso-structure in Section 3.3.1 shows, it is challenging to consider a unique model for each type. The scatter of each individual parameter, such as e.g., the crimp angle $\beta_{c}$ or the nesting factor $\eta_{P}$ (cf. Figure 9b,c) is too big that one unique model could be representative for the textile composite with a certain amount of layers.

Therefore two idealistic configurations (config.) are considered for each of the three textile types for a two layer stacking in the numerical determination of their fracture resistance at the desired combined load application ($\sigma_{w}^{t}/\sigma_{3}^{c}$). The two configurations are schematically illustrated in Figure 12. The no nesting (NN, rUC^(NN)^) in Figure 12a represents the most idealistic configuration of a textile reinforcement. All yarns are aligned equally and parallel in the weft and fill direction so that there is no superposition interlocking possible between each layer.

The second configuration—the maximum nesting (MN, rUC^(MN)^)—shown in Figure 12b, typifies the most possible interlocked configuration of each textile reinforcement. With that modelling approach, the fringes of the scatter should be represented. However, in a closer look of the theoretically and the experimentally determined nesting factors by Polturi
$\eta_{P}$, the values of the rUCs in the MN config. are more correlated to the experimental mean values of each textile reinforcement (cf. Table 3 and Table 5 and Figure 9c). This is apparently explained by the negligence of a deviation from the ideal yarn geometry, such as the yarn’s course or cross-section (cf. Figure 13a). Micrographs of type I and II, for example, showed a higher deformation of the yarns from the idealised geometry than type III. This evidence correlates to a higher deviation from the theoretical values of $\eta_{P}$ to the experimental (e.g., type I 0.831/rUC${}_{I}^{\mathrm{MN}}=\;$0.851 and type III 0.868/rUC${}_{\mathrm{III}}^{\mathrm{MN}}=\;$0.870). Nevertheless, both commonly used modelling software, such as TexGen [33] and WiseTex [32] which provide only the possibility of modelling idealised geometries. Further methods are to be considered, such as those presented in [17,64].

The modelling of the rUCs was governed by the open-source software TexGen from the University of Nottingham [33]. The software provides a convenient, beneficial and fast modelling of the textile geometry by full integration of its Python library in a graphical user-interface and export capability to export the geometry or mesh to Abaqus. The parameters for each type are obtained by the presented method in Section 3.3.1 and the used mean values are listed in Table 5. The sinusoidal shape of the yarns’ course are modelled by a spline approximation. The software thereby supports a huge variety, sucha s cubic Bezier splines, piece-wise polynominal splines, natural cubic splines and periodic cubic splines [65]. For the approximation of the yarns’ cross section, several mathematical descriptions— elliptical, lenticular, power ellipse (specialisation of the Lame’ian superellipse with $|\frac{y}{d}\left|{}^{n}+\right|\frac{x}{D}{|}^{n}=1$) or hybrid cross-section—are available for modelling [33]. In good agreement to the results of the image anlaysis in Section 3.3.1, the power ellipse was chosen with the minor $\bar{d}$ and major $\bar{D}$ diameter, according to Table 5 and the power factor n=1.8.

#### 4.1.2. Mesh and Boundary Conditions

Meshing the complex geometry of textile rUCs is quite challenging (cf. rUC${}_{I-\mathrm{III}}^{\mathrm{MN}}$ in Figure 13b–d). A conventional continuous meshing method, where the yarns and the matrix are meshed as one part (cf. Figure 14b is exemplary for an rUC of one filament and matrix), difficulties will ensue according to mesh quality and requirements on computational resources [35]. The embedded element method (EEM), on the contrary, provides the possibility to improve the mesh quality by using two independent superpositioned meshes. The two meshes are distinguished in the embedded part, the reinforcement, and in the host part, the matrix (Figure 14c). Both meshes are interconnected by their translational degrees of freedom using weighted average function [35,66]. All parts have been meshed using Abaqus and C3D8 and C3D8R elements for the embedded and the host parts, respectively. The mesh size was set identical to 0.100 mm for all rUCs. Any significant influence of the mesh size on the obtained results was not found. Necessarily, contact relations between the yarns were implemented to prevent the penetration of the meshes due to large deformations [63].

Other than in [44] the rUCs are subjected to Dirichlet boundary conditions (BC). The forces are derived from the experimental data and applied on the eight master nodes of each rUC, illustrated in Figure 13b–d, for the Abaqus|standard (implicit) solver. For any parallelepiped rUC, the application of periodic boundary condition (PBC), in particular 3D periodic boundary conditions, on the boundaries ($j^{+}$ and $j^{-}$) of the rUC ∂V is mandatory [67]. Figure 14a illustrates, schematically, an undeformed and deformed rUC considering PBC as exemplified by a two-dimensional filament-matrix material (Figure 14b,c). According to [67], the constrained periodic conditions on ∂V can generally be read to
(4)$$\begin{array}{cc} u_{i} & ={\bar{\varepsilon }}_{ik}x_{k}+u_{i}^{*}\;\mathrm{with}\;u_{i}\;\mathrm{periodic} \end{array}$$
with $u_{i}$ the displacement of each boundary, its periodic part $u_{i}^{*}$ and the average strain ${\bar{\varepsilon }}_{ik}$ on each boundary *j*. Hence for the rUC in Figure 14a the displacements in Equation (Equation 4) on a pair of opposite boundary surface can be derived to:
(5)$$\begin{array}{cc} u_{i}^{j+} & ={\bar{\varepsilon }}_{ik}x_{k}^{j+}+u_{i}^{*} \end{array}$$
(6)$$\begin{array}{cc} u_{i}^{j-} & ={\bar{\varepsilon }}_{ik}x_{k}^{j-}+u_{i}^{*} \end{array}$$


For any parallelepiped rUC, the difference of the displacements on each pair of opposite boundaries should be equal to zero despite the periodic displacements $u_{i}^{*}$. Consequently, the difference in Equations (Equation 5) and (6) is:(7)$$\begin{array}{c} u_{i}^{j+}-u_{i}^{j-}=c_{i}^{j}\;\left(\forall \;i,j=1,2,3\right) \end{array}$$

Thereby, for i=j the constant $c_{i}^{j}$ represents the average displacement in the normal directions 1, 2 and 3. For i≠j, the constants correspond to a shear deformation according to the three shear traction components (12, 13 and 23) [67].

#### 4.1.3. Elasticity

Describing the elastic properties of an rUC, that incorporates the superposition of meshes by the EEM, needs some modifications. Considering a volume ∂V of the host part that is fully super-positioned by an embedded part, it can schematically be described as two parallel springs, with the stiffness $C_{ijkl}^{\left(E\right)}$ and $C_{ijkl}^{\left(H\right)}$ for the embedded (E) and the host (H) part, respectively. Nevertheless, using the original obtained elastic values, the effective stiffness of the considered volume ∂V would be over estimated or spurious. This is known as volume redundancy and is comprehensively described in [68]. Regarding the volume redundancy, the use of the reduced stiffness for the embedded part $C_{{ijkl}_{R}}^{\left(E\right)}$ is mandatory. It reads to
(8)$$\begin{array}{cc} C_{{ijkl}_{R}}^{\left(E\right)} & =C_{ijkl}^{\left(E\right)}-C_{ijkl}^{\left(H\right)} \end{array}$$

Hence, the effective stiffness of the rUC reads to:(9)$$\begin{array}{cc} C_{ijkl}^{\left(rUC\right)} & =V^{\left(E\right)}\;\left(C_{{ijkl}_{R}}^{\left(E\right)}+C_{ijkl}^{\left(H\right)}\right)+V^{\left(H\right)}C_{ijkl}^{\left(H\right)} \end{array}$$
which has been incorporated in the numerical computation.

Regarding the stiffness of the host part $C_{ijkl}^{\left(H\right)}$, it was modelled by an elastic–plastic isotropic approach. The instaneous elastic properties $E_{m}$ and $\nu_{m}$ as well as the derived shear modulus $G_{m}$ are listed in Table 6. The elastic–plastic behaviour of RTM6-2 from Hexcel was described by an isotropic hardening behaviour in Abaqus|standard (*ELASTIC, TYPE = ISO, *PLASTIC, HARDENING = ISOTROPIC) [69]. The underlying stress–strain curve is illustrated in Figure 15. Thereby the yield strength $R_{p0.2}^{t}$ and the ultimate tensile strength $R_{m}^{t}$ are highlighted in the experimentally obtained data.

The stiffness of the embedded part $C_{ijkl}^{\left(E\right)}$, thus the reinforcement, is described as a transversely isotropic material, such as a curved UD-layer with a locally distributed material CS (cf. Figure 1). As mentioned in Section 3.3.2 the stiffness of an UD-layer is highly correlated to its inherit FVC. Thereby, semi-empirical analytical closed form solutions, presented in [50], have been used to obtain the five independent elastic properties of a UD-layer in dependence of the FVC. Table 7 lists the used values for the stiffness $C_{ijkl}^{\left(E\right)}$ at a $\bar{\phi }$ = 72%. This value of the FVC was obtained in Section 3.3.2 and is listed in Table 4. Additionally, the equations used in the micro-mechanical approach are indicated and referred to in Section B.1.

### 4.2. Implementation of Cuntze’s Failure Mode Concept

To determine the complex damage and failure behaviour on the meso scale, the Cuntze’s Failure Mode Concept (FMC) was suitable, because of its predictive capability for UD-layers under a superimposed static triaxial stress state [42,70]. The changes in [39] for the world-wide failure exercise II (WWFE-II 3D validation), where it scored very well, were regarding simplifying and reducing one model parameter, as well as by-passing numerical problems [71]. Furthermore, it offers the possibility to distinguish the complex fracture behaviour of the UD-layer, so that profound knowledge can be derived in the following numerical investigations of the rUCs on the meso scale.

According to Cuntze, there are five failure modes (FM) considerable that indicate the failure of a transversely isotropic brittle material. In theory, the fracture that leads to each specific FM is understood as a separation of a damage-free idealised material [38]. Figure 16 illustrates these different FM, which can be distinguished in two fibre failures (FFs) (Figure 16a,b) and three inter-fibre failures (IFFs) (Figure 16c–e) modes. Hereby an IFF indicates the onset of failure from the damage of the brittle material. In the case of the transversely isotropic description of the material rotation about the ‖|1-axis is redundant [39]. The three IFFs are allocated to a transverse tension stress $\sigma_{⊥}^{t}|\sigma_{2}^{t}$ (IFF1 $F_{⊥}^{\sigma }$, Figure 16c), a transverse compression stress $\sigma_{⊥}^{c}|\sigma_{2}^{c}$ (IFF2 $F_{⊥}^{\tau }$, Figure 16d) and an in-plane shear stress $\sigma_{⊥\|}|\sigma_{21}$ (IFF3 $F_{⊥\tau }$, Figure 16d). Thereby the superscripts σ and τ denote the failure mechanisms of each FM, whether it is driven by a normal fracture (σ) or a shear fracture (τ). The subscripts ⊥ and ‖ indicate the acting stress. For instance IFF2, is caused by a transverse compression stress $\sigma_{⊥}^{c}|\sigma_{2}^{c}$, which is a normal stress in terms of elasticity, but is allocated to a pure shear dominant fracture behaviour ($F_{⊥}^{\tau }$). For a detailed analytical explanation, it is recommendable to see Puck’s description of IFF-criteria in [57], which are based on a proposal by Hashin in the year 1980 [39]. Lastly, the two FFs are to be mentioned, which are caused by a fibre parallel compression $\sigma_{1}^{t}|\sigma_{\|}^{t}$ or tension stress $\sigma_{1}^{c}|\sigma_{\|}^{c}$. The two FMs are depicted in Figure 16a,b. Contrary to an IFF, an FF of a brittle material has to be assigned to a final failure that occurs spontaneously. Consequently, this indicates the loss of integrity of the hole material and is to be avoided regarding a reliable design with Cuntze’s failure mode concept (FMC).

Cuntze’s FMC is based on formulation by invariants. The advantage of using invariants for a transversely isotropic brittle material is the independence to a transformation of the CS [38]. According to the notation of the CS on micro scale in Figure 1, there are five independent invariants that can describe the multi-axial behaviour of the material. By the various formulations in the literature—e.g., $I_{3}^{B\mathrm{OEHLER}}=I_{4}^{H\mathrm{ASHIN}}$ and $I_{5}^{H\mathrm{ASHIN}}=-\sigma_{2}\tau_{31}^{2}-\sigma_{3}\tau_{21}^{2}+2\tau_{23}\tau_{31}\tau_{21}$ the following description will be used according to Cuntze [38]:(10)$$\begin{array}{cc} I_{1} & =\sigma_{1}\;\left(B\mathrm{OEHLER}\right) \end{array}$$
(11)$$\begin{array}{cc} I_{2} & =\sigma_{2}+\sigma_{3} \end{array}$$
(12)$$\begin{array}{cc} I_{3} & =\tau_{31}^{2}+\tau_{21}^{2} \end{array}$$
(13)$$\begin{array}{cc} I_{4} & ={\left(\sigma_{2}-\sigma_{3}\right)}^{2}+4\tau_{23}^{2} \end{array}$$
(14)$$\begin{array}{cc} I_{5} & =\left(\sigma_{2}-\sigma_{3}\right)\;\left(\tau_{31}^{2}-\tau_{21}^{2}\right)-4\tau_{23}\tau_{31}\tau_{21} \end{array}$$

Based on the assumption of the volume and shape change, according to a strain energy basis, the five different failure modes and their corresponding strengths read to [39,71]: (15)FF1:F‖⊥σ=I1R¯‖t=1(16)IFF1:F⊥‖σ=I2+I42R¯⊥t=1
(17)FF2:F‖⊥τ=−I1R¯‖c=1
(18)IFF2:F⊥‖τ=b⊥τ−1I2R¯⊥c+b⊥τI4R¯⊥c=1
(19)$$\begin{array}{cc} \mathrm{IFF}3:\;F_{⊥\|} & ={\left(\frac{\sqrt{I_{3}}}{{\bar{R}}_{⊥\|}}\right)}^{3}+b_{⊥\|}\frac{I_{2}\cdot I_{3}-I_{5}}{{\left({\bar{R}}_{⊥\|}\right)}^{3}}=1 \end{array}$$
with
(20)b⊥‖=R¯⊥‖⊥4−τ21⊥‖42σ2c⊥‖τ21⊥‖2R¯⊥‖
and
(21)b⊥‖τ=1+σ2cτ+σ3cτ/R¯⊥cσ2cτ+σ3cτ/R¯⊥c+σ2cτ−σ3cτ2/R¯⊥c

Thereby, $\bar{R}$ denotes the average strength of the material with its subscripts ‖ and ⊥ for the acting plane according to Figure 1 and its superscript *c* and *t*, classifying in a tension or compression stress. The so called interaction coefficients $b_{⊥\|}$ and $b_{⊥}^{\tau }$ are to be determined by the calibration points $\tau_{21}^{⊥\|}$, $\sigma_{2}^{c⊥\|}$, $\sigma_{2}^{c\tau }$ and $\sigma_{3}^{c\tau }$. For a detailed description of the determination of that calibration point, see [39,42,71]. Table 8 lists the five essential uniaxal strengths that were used in the numerical implementation of the FMC. The strengths are approximated to an FVC of 72%, according to the experimentally obtained ${\bar{\phi }}^{\left(m\right)}$ in Section 3.3.2. The values are in good accordance to the experimental data reported in [71].

The addressed simplification from the original description carried out in [38] to the revised formulations in [39] was regarding the interaction parameters $b_{⊥\|}$ and $b_{⊥}^{\tau }$ [71]. These parameters are the so-called friction-related model parameters based on a Mohr-Coulomb approach [39]. Triaxial load—in particular compressive load or hydrostatic pressure—have several phenomenological effects on the apparent fracture resistance of a composite material. Firstly, they can increase the damage tolerance by ‘*healing flaws*’, which corresponds to a higher stiffness or fracture resistance. This effect is particularly unspecified by the FMC. Secondly, they can increase the fracture resistance by elasto-mechanically ‘*strengthing*’ the compressed UD-material (Figure 17), which is described by the mentioned Mohr-Coulomb approach and allocated to the internal friction. This is an important aspect in the following consideration of the marco-scopic load combination ($\sigma_{w}^{t}/\sigma_{3}^{c}$) [39]. Determining these friction-based parameters is challenging, but is comprehensively described in [42,71]. In [38], empirical ranges for the parameters are specified for the desired material.

To describe the multidimensional fracture surface and the interaction between each FM, Cuntze reports an approach of a simple probabilistically based ‘*series spring model*’ [39]. Incorporating the stress efforts EFF and their corresponding equivalent stresses $\hat{\sigma }$ rather than the FM, according to Equations (Equation 15)–(Equation 19), detailed descriptions are presented in [38,39]; the resultant stress efforts can be derived to:(22)$$EFF^{{\left(res\right)}^{\dot{m}}}=\underset{EFF_{\|}^{\sigma }}{\underset{︸}{{\left(\frac{{\hat{\sigma }}_{\|}^{\sigma }}{{\bar{R}}_{\|}^{t}}\right)}^{\dot{m}}}}+\underset{EFF_{\|}^{\tau }}{\underset{︸}{{\left(\frac{{\hat{\sigma }}_{\|}^{\tau }}{{\bar{R}}_{\|}^{c}}\right)}^{\dot{m}}}}+\underset{EFF_{⊥}^{\sigma }}{\underset{︸}{{\left(\frac{{\hat{\sigma }}_{⊥}^{\sigma }}{{\bar{R}}_{⊥}^{t}}\right)}^{\dot{m}}}}+\underset{EFF_{⊥}^{\tau }}{\underset{︸}{{\left(\frac{{\hat{\sigma }}_{⊥}^{\tau }}{{\bar{R}}_{⊥}^{c}}\right)}^{\dot{m}}}}+\underset{EFF_{⊥\|}}{\underset{︸}{{\left(\frac{{\hat{\sigma }}_{\|⊥}^{}}{{\bar{R}}_{⊥\|}}\right)}^{\dot{m}}}}$$

Thereby $\dot{m}$ denotes the probabilistic interaction coefficient. Profound empirical evidence showed that $2.5<\dot{m}<3$ for carbon fibre reinforced plastic is suitable [38]. A detailed description and the formulation of the stress efforts are presented in Section B.2.

### 4.3. Numerical Analysis and Results

Assessing the numerical results, the following trigger conditions for failure or damage of the rUC have been declared:(23)$$\begin{array}{cc} 1 & \leq \left\{\begin{array}{cc} EFF_{\|,\max}^{\sigma } & \mathrm{failure}\\ EFF_{\max}^{\left(res\right)} & \mathrm{damage} \end{array}\right. \end{array}$$

As mentioned before, the appearance of IFF is mostly linked to the occurrence of damage. Whereby the accumulation of IFF is progressive and not spontaneous as with an FF. Additionally, it can be assumed that the resultant stress effort is mostly governed by accumulation of FM regarding an IFF at a multi-axial load combination [44]. Hence if in any yarns’ defined section (cf. Figure 13a) $EFF^{\left(res\right)}$ exceeds 1, it is to be assumed that damage accumulation commences at the prescribed load ratio (${\tilde{\sigma }}_{L}$, cf. Equation (Equation 2)). On the contrary, if $EFF_{\|}^{\sigma }$, which indicates IFF1 (cf. Figure 16a), occurs, we can assumed the integrity of the rUC decays. Hence the failure of the rUC is obtained.

For clarity, four load ratios (${\tilde{\sigma }}_{L}$ 1:0, 1:1, 1:0.7 –for better comparability to the experimentally obtained data, ${\tilde{\sigma }}_{L}=1:0.7$ has additionally been chosen– and 0:1) for the six considered rUCs (cf. Table 5) have been selected for the numerical assessment of the failure and damage behaviour of the three chosen textile reinforcements. For comparability and clarity of the load combination, as well as to avoid three dimensional plots, the following norm of numerical load vector has been defined, which actually resembles the von Mises stress of the homogenised remote stresses $\langle \sigma_{w}^{t}\rangle$ and $\langle \sigma_{3}^{c}\rangle$:(24)$$\begin{array}{cc} \left|\{\sigma_{L}^{\left(\mathrm{num}\right)}\}\right| & =|{\left[\langle \sigma_{w}^{t}\rangle ,\langle \sigma_{3}^{c}\rangle \right]}^{⊤}|\\ \; & =\sqrt{{\langle \sigma_{w}^{t}\rangle }^{2}+{\langle \sigma_{3}^{c}\rangle }^{2}} \end{array}$$

After validating the numerically identified strengths and fracture resistances for combined in-plane and out-of-plane loadings (cf. Section 4.3.1), respectively, within the experimentally obtained data (cf. Section 4.3.2), an advanced methodology was used to identify the influence of the fabric architecture, such as ondulation (cf. Section 4.3.3) and the configuration, such as nesting effects (cf. Section 4.3.4). Hereby, box plots (cf. Figure A2) are incorporated in a profound analysis approach to determine the individual effects on the damage and failure behaviour of the three textile reinforcements. On the contrary to the illustration of, for example, numerically computed stress effort, according to Cuntze, in three dimensional plots, box plots provide a more comprehensive insight into the distribution of the considered data in specified sections. Box plots graphically indicate representative values of a considered normally distributed value. Hence, rather a quantitative than a qualitative analysis can be obtained. A detailed description of the graphical notation of a box plot is provided in Section B.3.

#### 4.3.1. Analysis of Damage Initiation and Failure

Figure 18 illustrates the maximum resultant stress effort $EFF_{\max}^{\left(res\right)}$ according to Equation (Equation 22) in dependence on the norm of the numerical load vector $\left|\{\sigma_{L}^{\left(\mathrm{num}\right)}\}\right|$ (Equation (Equation 24)) as described before. Hereby the results for the load ratios $\left\{{\tilde{\sigma }}_{L}\right\}$ of 1:0 (Figure 18a), 0:1 (Figure 18b), 1:1 (Figure 18c) and 1:0.7 (Figure 18d) are selected for all six rUCs. All discrete values are listed in Table A3. Additionally, the threshold to distinguish between nondamaged state (<1) and damage initiation (>1) is marked by a red line.

Figure 18a,b illustrate the results for the uniaxial load path. In comparison to a uniaxial tension load, all rUCs under an out-of-plane compressive loading show a substantially earlier damage initiation ($EFF_{\max}^{\left(res\right)}>1$) but a lower influence on the fabric architecture and configuration regarding the lower scatter between the rUCs. For both load paths, rUC${}_{I}^{\mathrm{NN}/\mathrm{MN}}$ and rUC${}_{\mathrm{II}}^{\mathrm{NN}/\mathrm{MN}}$, an identical behaviour is observed. rUC${}_{\mathrm{III}}^{\mathrm{NN}}$ and rUC${}_{\mathrm{III}}^{\mathrm{NN}}$ have a much earlier damage evaluation for both load paths regarding the highest ondulation of the selected textile reinforcements. It is noticeable that there is a high influence of the configuration of each rUC in dependence of the load path, while every rUC in MN config., for the load ratio 1:0 ($\sigma_{w}^{t}$), indicates a higher fracture resistance regarding the damage initiation criterion (cf. Figure 18a), the opposite behaviour for the load ratio 0:1 (uniaxial $\sigma_{3}^{c}$) can be found in Figure 18b. For the combined load paths in Figure 18c (1:1) and Figure 18d (1:0.7) the results show similarity to the perceptions that could be derived from the uniaxial load combinations. The scatter is lower than for a uniaxial tension load $\sigma_{w}^{t}$ and by any prevalence of a $\sigma_{3}^{c}$-stress the MN configurations indicate a lower fracture resistance against damage.

Figure 19 illustrates the results for the determination of each rUCs’ failure according to Equation (Equation 23). This indicates the loss of the integrity and corresponds to a numerically obtained strength and fracture resistance for combined load paths, respectively, of the rUCs. In comparison to the linearly increasing maximal resultant stress effort $EFF_{\max}^{\left(res\right)}$, the maximum stress effort for FF1 $EFF_{\|}^{\sigma }$ indicates a more progressively increasing behaviour, especially with a prevailing $\sigma_{3}^{c}$-stress (cf. Figure 19b). For rUC${}^{N}N_{I}$, no converging solution could be obtained for the out-of-plane compressive stress load path ($\left\{{\tilde{\sigma }}_{L}\right\}$ = 0:1) so that the last increment with a maximum stress effort for FF1 of $EFF_{\|,max}<0.830$ has been used for the determination of the numerical uniaxial strength $R_{3}^{c}$. Regarding failure at the uniaxial load paths, the rUCs showed a slightly different behaviour according to damage initiation. The scatter for both load paths is quite comparable and there is a higher fracture resistance for a $\sigma_{3}^{c}$-loading (cf. Figure 19a,b). However, they share the same observations regarding the rUC’s configuration that the NN config. has a higher fracture resistance than the MN config. for any rUC and vice versa for a prevailing $\sigma_{w}^{t}$-stress. It could be assumed that the rUCs${}^{\mathrm{MN}}$ have a higher FVC regarding the geometrical and numerical modelling assumptions, which could lead to a higher obtained fracture resistance by a prevailing $\sigma_{w}^{t}$-stress. The fracture resistances obtained for the combined load paths in Figure 19c,d indicate a narrowly distributed scatter. Especially for rUC${}_{I}^{\mathrm{NN}/\mathrm{MN}}$ and rUC${}_{I}^{\mathrm{NN}/\mathrm{MN}}$, the scatter is the lowest. For rUC${}_{\mathrm{III}}^{\mathrm{NN}}$ and rUC${}_{\mathrm{III}}^{\mathrm{MN}}$, lower values are obtained.

The shift to the lowest values by an increasing $\sigma_{3}^{c}$-influence is recognisable between the load paths $\left\{{\tilde{\sigma }}_{L}\right\}$ = 1:0.7 and $\left\{{\tilde{\sigma }}_{L}\right\}$ = 1:1. A more profound knowledge of the location of the maximum $EFF_{\|}^{\sigma }$ has to be obtained to assume that a $\sigma_{3}^{c}$-stress is compensated through a $\sigma_{\|}$-stress in the yarns concerning a higher ondulation, which leads to more increasing $EFF_{\|}^{\sigma }$ for the same numerical load vector $\left|\{\sigma_{L}^{\left(\mathrm{num}\right)}\}\right|$ (Equation (Equation 24)).

On the contrary, a load path of $\left\{{\tilde{\sigma }}_{L}\right\}$ = 1:0 (cf. Figure 19a) indicates the same behaviour that an increasing ondulation leads to a lower fracture resistance against a $\sigma_{w}^{t}$-stress. This is in good agreement with other common research [20,37]. Nevertheless, the identification of the fracture resistance is provided by the identification of the maximum stress effort $EFF_{\|}^{\sigma }$. A closer look at the distribution of these values for the load path 1:0 reveals that the scatter increases by an increasing ondulation (cf. Figure 20). In Figure 20a,b the stress effort is narrowly distributed in the weft yarns for the considered load path. By increasing ondulation, on the one hand the scatter increases, and on the other hand the maximum values for identifying the fracture resistance appears in the vicinity of the edges of the yarns’ cross sections. This evidence could lead to a misinterpretation of results by a high ondulation. However this can only be validated by the experimental comparison, which is provided in the following Section 4.3.2.

Concluding, it could be derived from the identification of the damage initiation and failure occurrence that an MN config. has a higher influence on the fracture resistance for a uniaxial tension stress and vice versa for an NN config. concerning a prevailing $\sigma_{3}^{c}$-load application. Furthermore a higher ondulation which corresponds in a change of the textile architecture as defined results for any load case in a lower damage initiation and fracture resistance. Nevertheless the distribution of the values should be further investigated in Section 4.3.3 and Section 4.3.4 to obtain a deep understanding ofthe damage and failure behaviour concerning the textile architecture and configuration.

#### 4.3.2. Comparison of Experimental and Numerical Results

To validate the proposed method for the identification of the failure behaviour of textile reinforced composites, the numerically obtained strengths and fracture resistances, respectively, are compared to the experimental data from Section 3.1. In Figure 21, the results are presented for each considered textile type, according to Table 1. Thereby, the individual configurations MN and NN are separately plotted and linked. The experimental results are given with their determined mean value and standard deviation (std.) for two uniaxial load paths and three combined load paths, as described in Section 3.3. All discrete numerically obtained homogenised stresses ($\langle \sigma_{w}^{t}\rangle$, $\langle \sigma_{3}^{c}\rangle$) for damage and failure are listed in Table A3. Additionally, the relative error of the numerical values to the experimental mean values are listed in Table A3, to increase the comparability.

There is a good agreement of the numerical results for type I (rel. error for load path (1:1) and type I: −5.00% (NN) and −2.62% (MN)). Only a higher deviation is recognisable for a load ratio ${\tilde{\sigma }}_{L}$ of 1:3, which may correspond to computational uncertainties (Figure 21a). The relative error for load path 0:1, out-of-palne compression, and Type I is 7.81% (NN) and 8.23% (MN). The results for the uniaxial out-of-plane compression load path of rUC are underestimated, because it did not achieve the criterion for failure according to Equation (Equation 23). The maximum stress effort $EFF_{\|}^{\sigma }$ reached the value of 0.830 at a $\langle \sigma_{3}^{c}\rangle$-stress of −1035 MPa, as indicated in Figure 19b and Figure 21a and Table A3. For type II, there is still a good agreement with the experimental results (Figure 21b), especially for a prevailing $\sigma_{w}^{t}$-influence (load path 1:1 and type II: −0.24% (NN) and 10.22% (MN))). Numerically predicted results exceeding load paths of 1:2 overestimate the fracture resistance for type II. For type I this evidence can only be estimated due to the aforementioned reasons. Type III on the contrary underestimates the experimentally obtained values for all selected load paths. For load path 1:0 rel. error of −22.06% (NN) and −7.35% (MN) are determined, respectively. The decline in the predicted fracture resistance from load path 1:0.7 to 1:0 is not in accordance with the other results. It can be found that, for the out-of-plane load path, the rel. error of type III increases to −39.67% (NN) and −26.49% (MN), respectively. It may be related to the aforementioned findings that the stress effort of the rUC${}_{I}II^{\mathrm{NN}/\mathrm{MN}}$ to predict the failure according to Equation (Equation 23) has a higher scatter, and a maximum value of $EFF_{\|}^{\sigma }$ can be found in the vicinity of the yarns’ edges. This effect may be compensated by a superposition of a $\sigma_{3}^{c}$-stress.

As derived from the analysis, before the shift between the fracture resistance of the NN and MN configs. by a prevailing $\sigma_{3}^{c}$-stress influence and vice versa for a $\sigma_{w}^{t}$-stress is in evidence for all textile reinforcement types (cf. Figure 21a–c). The shift is in the range between the load paths 1:2 and 1:5. As stated in [17], nesting considerations are necessary to numerically predict the damage and fracture resistances of real textile reinforcements with their inherit nesting behaviour. To achieve a more profound understanding of the mechanical effects an advanced methodology with regard to the textile architecture and configuration will be incorporated and provided in the next Sections.

#### 4.3.3. Identification of the Influence of the Textile Architecture (Ondulation)

The following section shows the influence of the textile architecture by a probabilistic approach. Hereby the distribution in each yarn’s cross section will be assessed by box plots to quantify the distribution in each defined section along the yarns’ course (cf. Figure 14a). Figure 22 exemplarily illustrates the rUC${}_{I}^{\mathrm{NN}}$ and highlights the assessed symmetry volume. For the identification of the textile architecture such as ondulation, only the rUCs in the NN config. are used, because the influence of nesting effects can be neglected. Hence, the emphasis of the study is on the geometry of the yarn.

In the following section, the distribution of the damage initiation ($EFF_{\max}^{\left(res\right)}>1$, Equation (Equation 23)) and failure ($EFF_{\|,\max}^{\sigma }>1$, Equation (Equation 23)) will be assessed by box plots (cf. Figure A2). Additionally, for the damage initiation the composition of the resultant stress effort by each individual FM according to Cuntze is assessed and graphically illustrated by stacked bar plots. Thereby for comparability the fraction of each FM on the median value of $EFF^{\left(res\right)}$ in each individual section is derived by the following equation:(25)$$\begin{array}{cc} \eta_{EFF_{i}^{j}} & =\frac{{\left(EFF_{i}^{j}\right)}^{\dot{m}}}{\sum \left({\left(EFF_{i}^{j}\right)}^{\dot{m}}\right)}=\frac{{\left(EFF_{i}^{j}\right)}^{\dot{m}}}{\sum {\left(EFF^{\left(res\right)}\right)}^{\dot{m}}}\;\mathrm{with}\;i=\left\|,⊥,\right\|⊥\;\mathrm{and}\;j=\sigma ,\tau \end{array}$$

• Damage ($EFF_{\max}^{\left(res\right)}>1$, Equation (Equation 23)).

The results of the distribution of the damage initiation for the uniaxial load path 1:0 and each type are displayed in Figure 23a for the considered weft yarn sections and in Figure 23b for the considered fill yarn sections. For comparability, the values are plotted over the half nominal ondulation length x/λ and y/λ, respectively. From the results graphically illustrated in Figure 23. it can definitely be derived that for a uniaxial tension load in weft direction, damage will occur first of all in the fill yarn.

Values of $EFF^{\left(res\right)}$ exceeding the threshold value of 1 can be found in the sections of the fill yarn (Figure 23b). Thereby the scatter in each section increases from type I to III for both considered yarns and a range of $EFF^{\left(res\right)}$ can be identified that varies from approx. 0.6 to 1 for type I, 0.7 to 1 for type II and 0.45 to 1 for type III. The influence of the ondulation is recognisable by the increase and decrease in the obtained median values of $EFF^{\left(res\right)}$ over one half ondulation length. Thereby the highest values are computed in the middle section (y/λ=0.2–0.3).

On the contrary, the range of median values for the weft direction varies between approximately 0.2 to 0.5 and the values of each section are narrowly distributed. In comparison to the distribution of the fill yarn, there is a significant difference. Furthermore, a significant difference is in the composition of the individual FM, according to Cuntze’s FMC (cf. Figure 16). In Figure 24, the fractions $\eta_{EFF_{i}^{j}}$ for each type are plotted over the normalised ondulation length for each considered section and yarn as a stacked bar plot.

Whereby the weft yarn is significantly dominated by FF1, according to Equation 16, the resultant stress effort for the fill yarn is purely accumulated by IFF1. The fraction of $F_{⊥}^{\sigma }$ decreases by an increasing ondulation angle in the weft yarn for a uniaxial tension load $\sigma_{w}^{t}$ (Figure 24a). Theoretically, it should be the inverse due to the higher load deflection caused by the ondulation, especially in Section 1 and Section 5 (x/λ=0–0.01 and 0.40–0.50) (see also Figure 14a).

For the combined load case, $\{{\tilde{\sigma }}_{L}\}=1:1$ the difference in the distribution of $EFF^{\left(res\right)}$ of the weft and fill yarn is negligible. Both yarns for type I and II indicate $EFF^{\left(res\right)}$-values in the interquartile range (IQR) (cf. Section B.3) in specific sections exceeding the threshold value of 1. On the contrary, type III surpasses the threshold by some outlines. This implies that, if type III meets the damage initiation criterion that it is not as affected by damage, then the other two are considered textile reinforcements. It can be assumed that a higher ondulation results in a more damage-tolerant behaviour. Additionally, in comparison of the median values of $EFF^{\left(res\right)}$ there is a significant difference from type I and II with approx. values accumulating in the range of 0.55 to 0.85 for the weft yarn and 0.30 to 0.90 for the fill yarn to type III with values in the range of 0.40 to 0.60 for the weft and 0.30 to 0.65 for the fill (Figure 25).

The highest scatter and median values of $EFF^{\left(res\right)}$ are located for all three textile reinforcements in the sections 2 to 4 ((x,y)/λ=0.10–0.40), especially for the weft yarn. In a closer look on Figure 25b, the median values increase significantly in that area. With additional knowledge about the composition of the FM from Figure 26b, it can be stated that there is a change in the mode interaction of the fill yarn. The fraction $\eta_{EFF_{⊥}^{\tau }}$, which corresponds to the IFF2 caused by transverse compression (cf. Figure 16d), is substantially higher in those sections for all types. On the contrary, the $\eta_{EFF_{⊥}^{\tau }}$ is quite constant with a minor decrease in the middle sections for the weft yarn (cf. Figure 26a). In comparison to the composition of the FMs for a uniaxial tension load (cf. Figure 24) to the load path $\{{\tilde{\sigma }}_{L}\}=1:1$, there is a major transition identifiable. Where pure $\sigma_{w}^{t}$ would cause IFF1 in the fill yarn (cf. Figure 24b). Only the minor superposition of $\sigma_{3}^{c}$-stress shifts the damage behaviour to a combination of IFF2 and IFF3 for both yarns (cf. Figure 26). Hence, it can be concluded that the superposition of a $\sigma_{3}^{c}$ has a major influence on the damage behaviour of a textile reinforced composite.

To conclude, the analysis of the influence of the textile architecture on the damage initiation, Figure 27 illustrates the distribution of $EFF^{\left(res\right)}$ at the identified damage initiation for a uniaxial compression load path ($\left\{{\tilde{\sigma }}_{L}\right\}$ = 0:1). The highest values can be obtained in the sections 2 to 4 ((x,y)/λ=0.10–0.40) in weft and fill direction. The medium values of $EFF^{\left(res\right)}$ vary between 0.40 to 1. The aforementioned findings that the median values for type III are the lowest ($EFF^{\left(res\right)}=0.40–0.65$) can be determined as well (cf. Figure 27). Consequently, the criteria according to Equation (Equation 23) was reached and triggered by some outlines, while for types I and II, a much higher damage state could be predicted.

The difference in the distribution of $EFF^{\left(res\right)}$ is marginal for both yarns (cf. Figure 27a,b). A pure $\sigma_{3}^{c}$ induces the same stress state in the weft and fill directions. Consequently, the same resultant stress effort and even the same composition of FM can be observed in Figure 28. The composition of FM is significantly governed by IFF2 ($\eta_{EFF_{⊥}^{\tau }}$) for both directions. There is even no major difference on the mode interaction due to an increasing ondulation, as the narrowly similar distribution of $\eta_{EFF_{i}^{j}}$ implies. In comparison to a uniaxial in-plane tension load ($\sigma_{w}^{t}$), where damage accumulates mainly in the fill yarn (cf. Figure 23), damage occurs simultaneously in the weft and fill yarn for a uniaxial out-of-plane compression load ($\sigma_{3}^{c}$).

• Failure ($EFF_{\|,\max}^{\sigma }>1$, Equation (Equation 23))

Referring to Section 4.3.2, the proposed method can sufficiently predict the fracture resistances for types I and II—the textile reinforcements with the lowest ondulation—by this numerical approach. Nevertheless, as qualitatively estimated for type III in Section 4.3.1 by Figure 20, it can quantitatively been validated by the assessment of the distribution by box plots (cf. Figure 29). While the median values of $EFF_{\|}^{\sigma }$, the stress effort for FF1 for types I and II, are narrowly distributed in the range of approx. 0.86 to 0.98, the median values of type III vary between 0.60 to 0.65. Values exceeding the threshold value of 1 are only present in the upper whisker or as outlines for all types. Hence the failure of type III is underestimated by the proposed method. Furthermore it can be assumed that failure behaviour is less spontaneous than for type I or II, because there is less strain-energy available causing a non-catastrophic loss of integrity. This evidence may correlate to the terminologically determined failure behaviour of type III, as described in Section 3.3. Unless the progressive damage evolution in particular delamination was not incorporated for the numerical analysis.

This abstained from illustrating the distribution of the fill yarns in Figure 29, because the stress effort $EFF_{\sigma }^{\|}$ in the fill yarn is negligible (cf. Figure 20). By the illustration of an entire ondulation length, the assumption of symmetry can additionally be confirmed.

On the contrary, a superposition by a $\sigma_{3}^{c}$-stress yields equal shares in fill and weft yarn for $\sigma_{\|}^{t}$-stress, due to the induced transverse strain. Therefore, the stress effort $EFF_{\|}^{\sigma }$ is illustrated for both yarns in Figure 30. It is clearly recognisable that, for a load ratio $\left\{{\tilde{\sigma }}_{L}\right\}$ = 1:1, the weft yarns are still the major stressed reinforcement. Hereby the same pattern is being emphasised, such that type I and II are narrowly distributed and the median of $EFF_{\|}^{\sigma }$ varies between 0.80 to 0.90, while type III has a significantly higher scatter and median values in the range of 0.50 to 0.55 (cf. Figure 30a). The values of the box plots in Figure 30b indicate that there is no failure in the fill yarn expectable. Despite the fact that $EFF_{max}^{\left(res\right)}$ of 0.32 have been induced by transverse strain, it is negligibly small compared to the stress effort in the weft. Hence, for a load ratio of 1:1, failure is mainly governed by the weft and damage is to be expected in the fill yarn.

On the contrary, for a uniaxial load path of 0:1 ($\sigma_{3}^{c}$), the failure can be expected in weft and fill direction, as Figure 31 indicates. The stress efforts $EFF_{\|}^{\sigma }$ are equally distributed in the fill and weft yarn (cf. Figure 31a,b). The values vary between the range of 0.6 to 0.8 of the median value of $EFF_{\|}^{\sigma }$ for types I and II at the predicted failure stress state according to the analysis method (cf. Figure 19b). Type III repeatedly shows far lower median values ($EFF_{\|}^{\sigma }=0.50–0.55$) and a higher scatter. Thereby it can be assumed that the failure criterion has been triggered by any outliners in the rUCs and the predicted strengths are underestimated. In comparison to the experimentally obtained strengths in the range of 907 to 960 MPa (cf. Figure 7 and Table A2), however, the numerical strengths are overestimated especially for types I and II, which vary between 1030 to 1121 MPa (cf. Figure 21 and Table A3). Concerning the fact that the predicted damage initiation occurs at load levels of $\langle \sigma_{3}^{c}\rangle =100–200$ MPa, it can be assumed that the failure behaviour of a real textile reinforcement would mainly be governed by accumulating damage of IFF2 caused by transverse stress ($\sigma_{⊥}$) in the yarns (cf. Figure 27 and Figure 28). This FM is, as described in Section 4.2, mainly driven by shear failure, which causes the typical inclined fracture plane, as illustrated in Figure 16d. These findings can sufficiently be related to the phenomenologically determined fracture pattern illustrated in Figure 6a for a uniaxial $\sigma_{3}^{c}$-load and in Figure 6c for a combined $\sigma_{w}^{t}/\sigma_{3}^{c}$-load with prevailing $\sigma_{3}^{c}$-superposition. Unfortunately, a progressive damage law has not been incorporated in the numerical evaluation of the considered rUCs and load paths.

#### 4.3.4. Identification of the Influence of the Textile Configuration (Nesting Effects)

As Figure 21 indicates, not only the textile architecture has an impact on the damage initiation and fracture resistance of a textile reinforced composite, but the textile configuration with different nesting effects has to be considered in any numerical approach. In the following section, the effects of nesting are considered either on the damage initiation ($EFF^{\left(res\right)}>1_{\max}$) or on the failure. To avoid redundant information and to increase clarity, the comparison to the NN config. is conducted on selected examples, and it is emphasised on the fill yarn, where damage initiation is most likely for any considered load combination. Hence, the emphasis on the evaluation is the composition of the resultant stress effort and distribution of its median values for the damage initiation and the distribution of $EFF_{\|}^{\sigma }$ illustrated by box plots for the failure of each rUC${}^{\mathrm{MN}}$.

• Damage ($EFF_{\max}^{\left(res\right)}>1$, Equation (Equation 23))

From the aforementioned findings, damage occurs mainly in the fill yarns for a load ratio of $\left\{{\tilde{\sigma }}_{L}\right\}$ = 1:0. Hence, for clarity, Figure 32 only illustrates the considered fill yarn. As for an rUC${}^{\mathrm{NN}}$, the damage initiation for rUC${}^{\mathrm{MN}}$ at the considered load path is principally triggered by IFF1. The values of its composition vary in the range of approx. 98%. However, in particular, a closer look at the median values of each configuration indicates that the influence of the inherent textile architecture decreases. The values of each rUC${}^{\mathrm{MN}}$ vary between 0.89 to 0.91, 0.82 to 0.85 and 0.64 to 0.71 for types I, II and III, respectively, whereby the rUCs${}^{\mathrm{MN}}$ have a much higher deviation and the influence of the ondulation is clearly emphasised (type I: 0.9 to 1; type II: 0.92 to 1; type III: 0.77 to 0.83). Obviously, the MN config. counteracts the effects of the specific textile ondulation.

This finding can also be perceived for a uniaxial out-of-plane compression load path ($\left\{{\tilde{\sigma }}_{L}\right\}$ = 0:1). As Figure 33 illustrates, the deviation of the median values of the resultant stress effort $EFF_{\mathrm{median}}^{\left(res\right)}$ of rUC${}^{\mathrm{NN}}$ is much higher than these for rUC${}^{\mathrm{MN}}$ at the same remote stress and homogenised stress, respectively ($\langle \sigma_{3}^{c}\rangle$).

Additionally, it can be derived from Figure 33, in comparison to Figure 28b, that not only the distribution of $EFF_{\mathrm{median}}^{\left(res\right)}$ is affected by the textile configuration, but even the composition $\eta_{EFF_{i}^{j}}$ of the individual FM in each section. Whereas the damage initiation is mainly triggered by the same shares of IFF2 and IFF3 ($\eta_{EFF_{⊥}^{\tau }}\approx 0.58$ and $\eta_{EFF_{⊥\|}}\approx 0.42$), there is a much higher deviation in the shares for the NN config. ($\eta_{EFF_{⊥}^{\tau }}\approx 0.55–0.65$; cf. Figure 28b). Hence, an MN config. weakens the effect of the textile architecture even on the the alternating FM. Concerning the experimental findings that the higher the ondulation the higher the nesting factor, which means less interlocking of adjacent layers (cf. Figure 9a,b), it can be derived that type I has a more evenly distributed damage occurrence than type III with the highest ondulation, crimp angle and nesting factor $\eta_{P}$, respectively.

• Failure ($EFF_{\|,\max}^{\sigma }>1$, Equation (Equation 23)).

As the comparison of the numerical and experimental results in Figure 21 unveiled that, for a uniaxial tension load path ($\left\{{\tilde{\sigma }}_{L}\right\}$ = 1:0), the MN config. shows a much higher fracture resistance than the NN config. and vice versa for a uniaxial out-of-plane compression load path ($\left\{{\tilde{\sigma }}_{L}\right\}$ = 0:1). To identify the influence of the textile, the configurations of both rUCs for each type are compared at the lower fracture resistance—this means, for load path 1:0, the load level of rUC${}^{\mathrm{NN}}$ and vice versa for load path 0:1. The goal is to identify any influential characteristics, as seen before on the damage behaviour—for example, a stiffer behaviour of the rUCs${}_{I-\mathrm{III}}^{\mathrm{MN}}$ that might lead to a higher fibre parallel tensile strength ($R_{w}^{t}$, cf. Figure 19a and Figure 21).

Figure 34 illustrates a full ondulation of a weft yarn and for each type its box plot per section. Thereby the coloured symbol represents the MN config. and the blank the corresponding NN config. Thereb,y the load levels are 770, 750 and 530 MPa for types I, II and III, respectively (cf. Table A3). The aforementioned influence of the MN config. on the distribution of the $EFF^{\left(res\right)}$ is not given. The influence of the ondulation is still emphasised by the alternating median values along the yarns’ courses. The highest values are mainly situated in section 3 and λ/2 later in section 8.

Nevertheless, it appears that the results from rUC${}_{\mathrm{III}}^{\mathrm{MN}}$ are more densely distributed than the results from rUC${}_{\mathrm{III}}^{\mathrm{NN}}$. This trend is valid for types I and II as well. Considering that all rUCs are loaded by Dirichlet-BC in particular forces, it can be assumed that the predicted stiffer behaviour of the rUC${}^{\mathrm{MN}}$, due to the higher FVC, affects only the homogenised strains and not the determined homogenised stresses $\langle \sigma_{ij}\rangle$. Hence, it can be derived from Figure 34 that a MN config. weakens the effects of the textile architecture. Additionally, this effect can also be graphically identified by comparing Figure 20c–f. Thereby a much more narrow distribution of $EFF_{\|}^{\sigma }$ can be found in the illustration, even at each determined failure state. That means that rUC${}_{\mathrm{III}}^{\mathrm{MN}}$ has an 18% higher $\langle \sigma_{w}^{t}\rangle$ load in Figure 20. In summary, an MN config. affects the stress distribution in the weft yarns positively—in particular, it decreases the scatter. Furthermore, it can be estimated by comparing the increase in the determined fracture resistances for $\left\{{\tilde{\sigma }}_{L}\right\}$ = 1:0, that the higher the ondulation the higher the increase (cf. Table A3, type I: 12%, type II: 22% and type III: 18%). However, on the contrary, considering the experimental results from Section 3.3.1, that a higher ondulation causes a lower nesting factor, this has to be incorporated by weighted functions to determine the fracture resistances for any textile reinforcement. Nevertheless, comparing the mean nesting factors determined by the presented numerical method in Table 3 to the values obtained from the rUC geometry in Table 5, it can be found that, for type III, $\eta_{P}$ is in good accordance (type III $\eta_{P}$: exp. 0.868; num. 0.870), but the values of rUC${}_{I-\mathrm{II}}^{\mathrm{MN}}$ indicate that there is a much higher nesting behaviour than the idealised geometry is capable to depict (type I $\eta_{P}$: exp. 0.831; num. 0.852; type II $\eta_{P}$: exp. 0.842; num. 0.874). Micrographs from types I and II showed that textile reinforcements with a lower areal weight deviate much more from the idealised yarn shape, such as the cross section, especially where nesting effects are present. These effects should be incorporated in further analyses.

On the contrary, nesting effects have a negative influence on the fracture resistance by prevalence of a $\sigma_{3}^{c}$-stress (cf. Figure 21). Despite the fact that a nesting configuration weakens the influence of the textile architecture on the damage behaviour, as illustrated in Figure 33, Figure 35 reveals a more distinctive behaviour. As before, the coloured box plots rely on the results of $EFF_{\|}^{\sigma }$ sections of each rUC${}^{\mathrm{MN}}$ and the corresponding rUC${}^{\mathrm{NN}}$ are blank for clarity. For comparability, the $\langle \sigma_{3}^{c}\rangle$-load for each type corresponds to the predicted failure load of the MN config., unlike the preceding illustration.

The main difference to the NN config. is that the peak values of $EFF_{\|}^{\sigma }$ are situated in the sections with the highest crimp (sections 1, 6 and 10). In this section, the interlocking of the adjacent layer is dominating. Concerning the idealised geometry of the configurations in Figure 12, a much stiffer area is located in the vicinity of these sections 1, 6 and 10. The fill yarn of the adjacent layer can be found here. It can be assumed that here a much higher load transmission of the out-of-plane compression stress than of the matrix-rich region of the NN config. that results in an transverse stress $\sigma_{⊥}$ in the weft yarn. This leads to a much higher stress effort $EFF_{\|}^{\sigma }$ in the crimp region by transverse strain. Nevertheless, significance can not be determined, which leads to the minor fracture resistance of the rUCs in the MN config. for an out-of-plane compression stress $\sigma_{3}^{c}$.

## 5. Conclusions

In this paper, advanced experimental and numerical methods are presented, which are capable of identifying the influence of textile reinforcement at combined in-plane and out-of-plane load on the failure and damage behaviour. To determine the influence of the ondulation and nesting three plain, weave fabrics with increasing areal weight and equal yarns have been considered. By increasing areal weight, the geometry of each yarn is affected by an higher ondulation, crimp angle, altered yarns’ cross section and a minor nesting capability, respectively.

For the experimental purpose, a new testing method was elaborated. This method provides the load combination of an in-plane fibre-parallel tensile stress and an out-of-plane compression stress by adaption to a planar biaxial testing machine. The pre-selected textile reinforcements have been tested to determine the fracture resistance at different combined load paths. Incorporating a contactless optical strain measurement system was suitable to sufficiently obtain the stress–strain relationship of each specimen. Additionally, standardised tests have been conducted to determine the uniaxial in-plane tensile strengths and the out-of-plane compressive strength. It could be derived that, especially for a prevailing in-plane tensile load, the influence on the fracture resistance of the textile reinforcement is significant. In particular, a higher areal weight and ondulation can even lead to an altered fracture and failure behaviour.

Advanced methods of computer vision and image processing have been utilised to experimentally examine the meso and micro structure of each textile reinforcement. From the analysis of acquired in situ CT data, the distribution of necessary parameters, such as ondulation, crimp angle and nesting factors, respectively, could sufficiently be determined (meso structure; textile architecture and configuration). Thereby, the aforementioned qualitative behaviour of the influence of the areal weight could be quantitatively validated. The fibre volume content was analysed using a new probabilistic method. Acquired micrographs have been analysed by an algorithm incorporating a Hough circle detection and a Delaunay triangulation.

This experimental data could sufficiently be used to derive idealised representative unit cells for the numerical approach. Thereby the two extreme textile configurations have been generated with the open source software TexGen. To obtain a good mesh quality, the method of embedded elements in the finite element solver Abaqus was sufficient. Incorporating Cuntze’s failure mode concept enabled the possibility to assess the failure and damage behaviour of the textile reinforcements. It was defined that, if the maximum resultant stress effort exceeds its limit, the occurrence of damage is to be presumed, and if the maximum stress effort for the fibre parallel failure mode exceeds its limit, the loss of integrity of the representative unit cell occurs, which means failure. In comparison to the experimentally determined fracture resistances and uniaxial strengths, repsectively, the following observations could be found:Textile reinforcements with minor areal weight;
-Good agreement to experiment for prevailing in-plane tensile stress;-Higher deviation to the experiment for the out-of-plane compression load path;Higher deviation (underestimation) from the experiment of the reinforcement with the highest areal weight for considered load paths;Nesting results in a higher fracture resistance for load paths with dominating in-plane tensile stress and in a lower fracture resistance for load paths with dominating out-of-plane compression stress compared to an idealised no nesting configuration.

An advanced statistical methodology was presented and discussed in this paper. It uses box plots to graphically illustrate and to compare the distributions of the stress efforts according to Cunze’s failure mode concept along selected yarns in predefined sections for each type. Regarding the textile architecture, it could be derived that a higher ondulation causes:A non homogeneous distribution of the stress efforts for all considered load paths, in particular a higher scatter. Hence less accuracy of the considered failure stress state (underestimation);Lower scatter of median values of the stress efforts for damage initiation and failure along the weft or fill yarn direction;Less influence of the ondulation on the distribution for a out-of-plane compression loading. The peaks alternating with the ondulation are decreased.

In comparison to each rUC in the two configurations, it can be stated that a nesting configuration causes

Less influence of the ondulation on the damage initiation;Altered load transmission that leads to shifted failure initiation for out-of-plane loading.

Summarising the experimental and numerical results, the following findings can be generalised that increasing areal weight leads to

Higher ondulation;Less nesting capability;More pronounced damage and failure behaviour;More damage tolerant behaviour and less spontaneous failure, which can be beneficial for impact or fatigue application;Less ultimate strengths for in-plane tensile load;Higher ultimate strengths for out-of-plane compression loading.

Concluding, it can be stated that these presented methods contribute to a more profound understanding of the influence on the damage and fracture resistance of textile reinforcements; in particular, of the ondulation and nesting behaviour of the damage behaviour and fracture resistance. Especially, improving the numerical methods can be beneficial for a better understanding of the damage and failure behaviours of textile reinforced composites. The incorporation of a progressive damage law, such as Böhm’s derivation of Cuntze’s failure mode concept, presented in [12], can be beneficial to more sufficiently approximate the ultimate fracture resistances. A more profound knowledge and incorporation of the distribution of the fibre volume content of each yarn can contribute to an improved approximation of the elastic properties and strengths on meso-scale. The incorporation of non-idealistic yarn geometries by a preceding simulation of the compaction behaviour of the dry layer, such as that presented in [17], may contribute to deeper insights about the nesting effects. Additionally, the incorporation of a semantic segmentation algorithm, by means of artificial intelligence, can benefit the determination of the yarn shape in image processing analysis. Nevertheless, further studies have to be conducted to contribute to design guidelines for advanced textile reinforced composite structures under combined in-plane and out-of-plane loading.

## 6. Patents

From previous work, which is included in this manuscript, the following German patent resulted: Andrich, Manuela; Hufenbach, Werner; Schirner, Robert: Verfahren zur Ermittlung werkstoffmechanischer Kennwerte an textilverstärkten Faser-Kunststoff-Verbunden. DE 102014203706.

## Figures and Tables

**Figure 1 materials-13-04772-f001:**
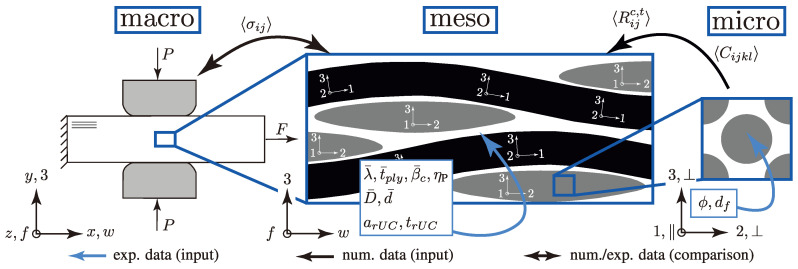
Schematic illustration of experimental and numerical multi-scale approach and notation of each used coordinate system.

**Figure 2 materials-13-04772-f002:**
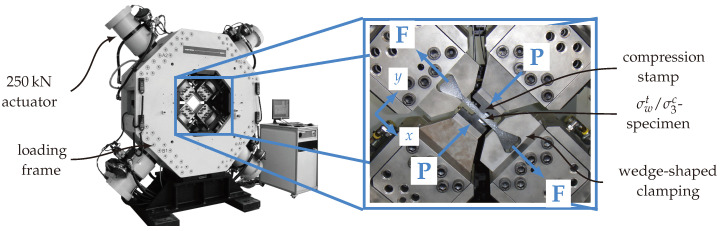
Planar biaxial servo-hydraulic testing machine Instron 8800 and magnification of the $\sigma_{w}^{t}/\sigma_{3}^{c}$-testing set-up.

**Figure 3 materials-13-04772-f003:**
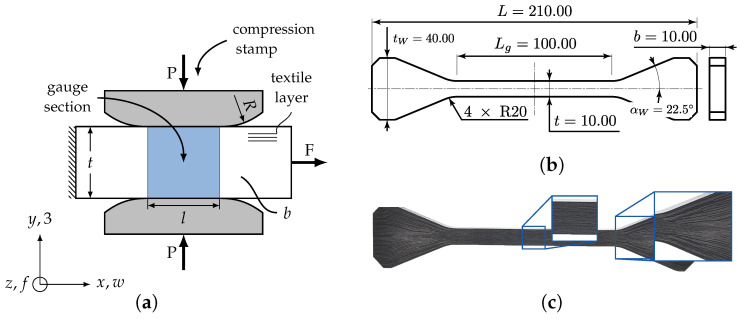
Elaborated $\left(\sigma_{w}^{t}/\sigma_{3}^{c}\right)$-testing for textile-reinforced composites: (**a**) schematic test set-up and global and material CS, (**b**) technical drawing with characteristic dimensions in (mm) and (**c**) specimen and magnification of wedge clamping section and gauge section.

**Figure 4 materials-13-04772-f004:**
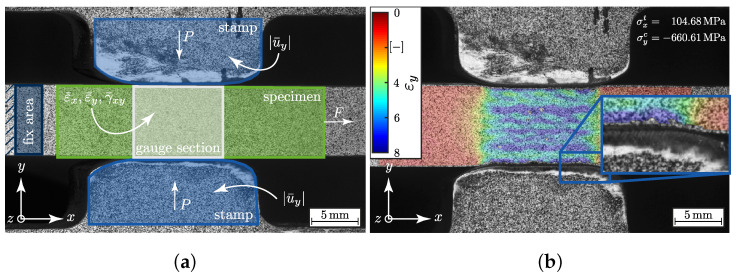
(**a**) Illustration and definition of evaluation areas of the 2D strain field measurement with DIC (Aramis^®^); (**b**) exemplary illustration of the obtained $\varepsilon_{y}$ strain field at a prescribed load path of $\left\{{\tilde{\sigma }}_{L}\right\}$ = 1:5 (Equation (2)) and magnification of the stamp indentation.

**Figure 5 materials-13-04772-f005:**
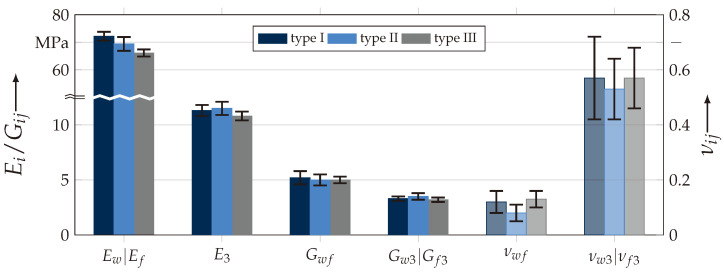
Illustration of elastic properties of ECC-style fabrics/RTM6-2 composite material ($\bar{\phi }\approx$ 60%; see Table A1 for explicit values).

**Figure 6 materials-13-04772-f006:**
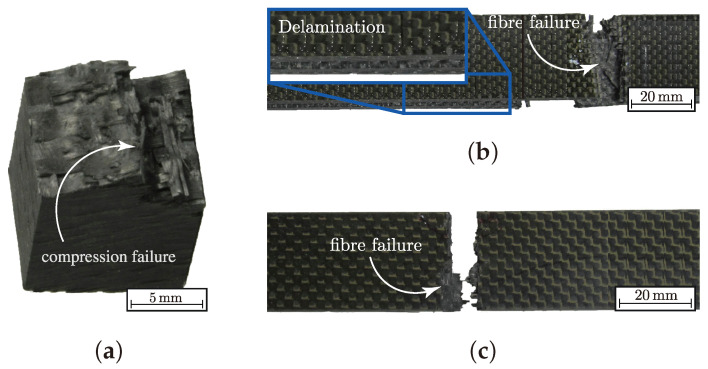
Phenomenological failure modes for different load paths: (**a**) compression failure ($\sigma_{3}^{c}$-testing [54]), (**b**) tension failure and (**c**) delamination failure (**b**,**c**): $\sigma_{w}^{t}$-testing [56]).

**Figure 7 materials-13-04772-f007:**
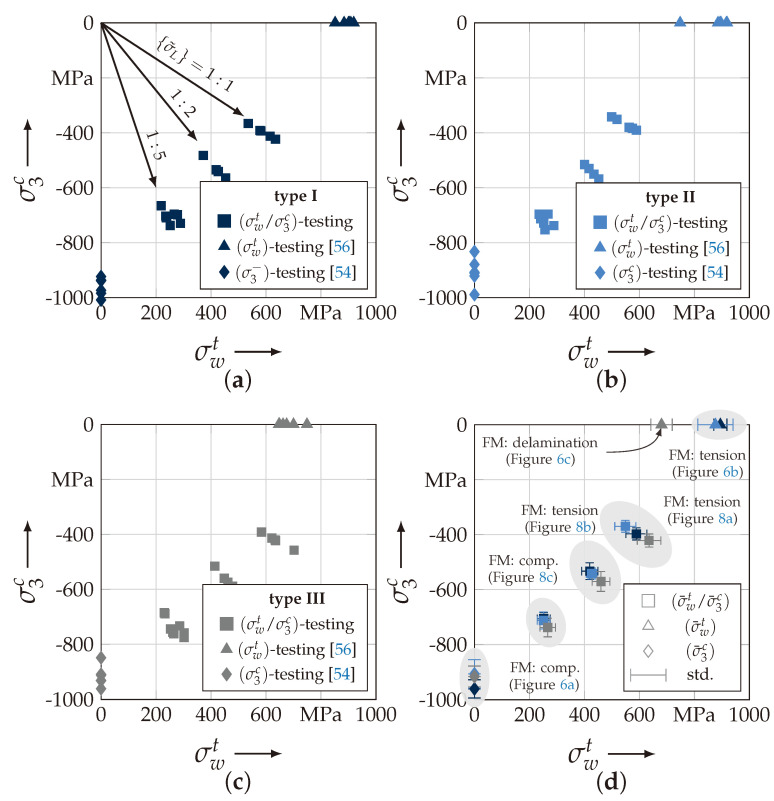
Experimental results: (**a**) type I, (**b**) type II, (**c**) type III (according to Table 1) and (**d**) comparison of mean values and classification of failure modes.

**Figure 8 materials-13-04772-f008:**
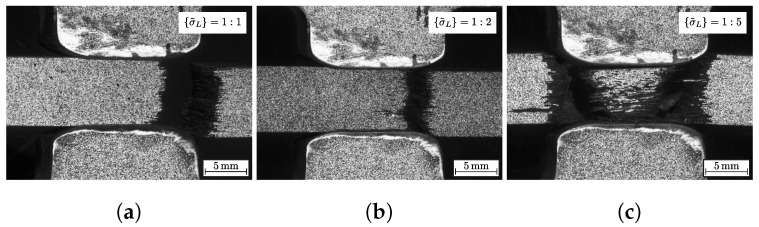
Phenomenological failure modes at different load paths of the $\left(\sigma_{w}^{t}/\sigma_{3}^{c}\right)$-testing: (**a**) tension failure (1:1), (**b**) tension failure (1:2), (**c**) compression failure (1:5).

**Figure 9 materials-13-04772-f009:**
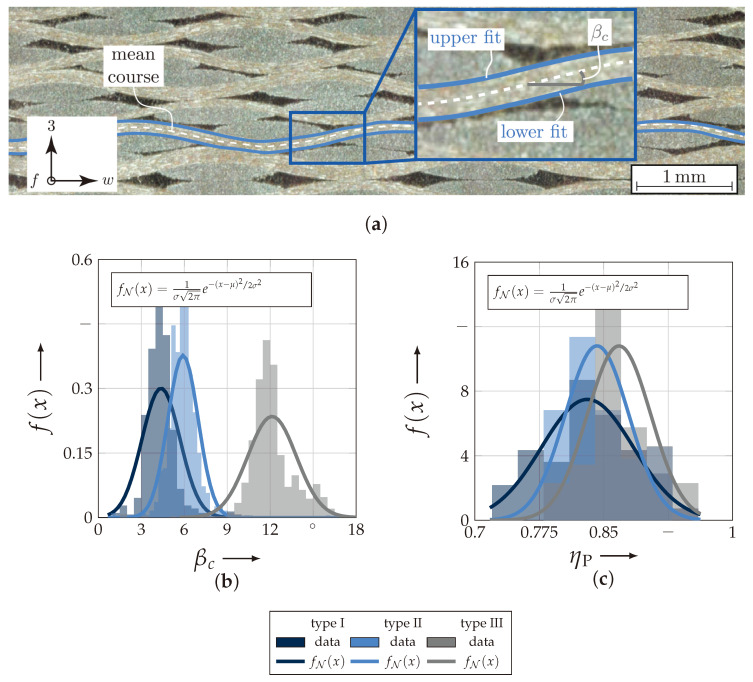
Illustration of (**a**) results of image processing method; histogram for each type and fit of normal probability density functions ($f_{N}\left(x\right)$) for (**b**) crimp angle $\beta_{c}$ and (**c**) nesting factor $\eta_{P}$.

**Figure 10 materials-13-04772-f010:**
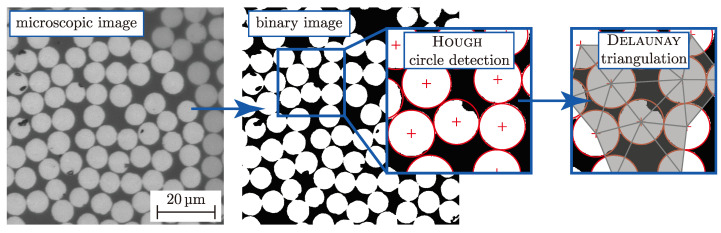
Schematic illustration of the proposed algorithm determining the microscopic FVC $\phi^{\left(m\right)}$.

**Figure 11 materials-13-04772-f011:**
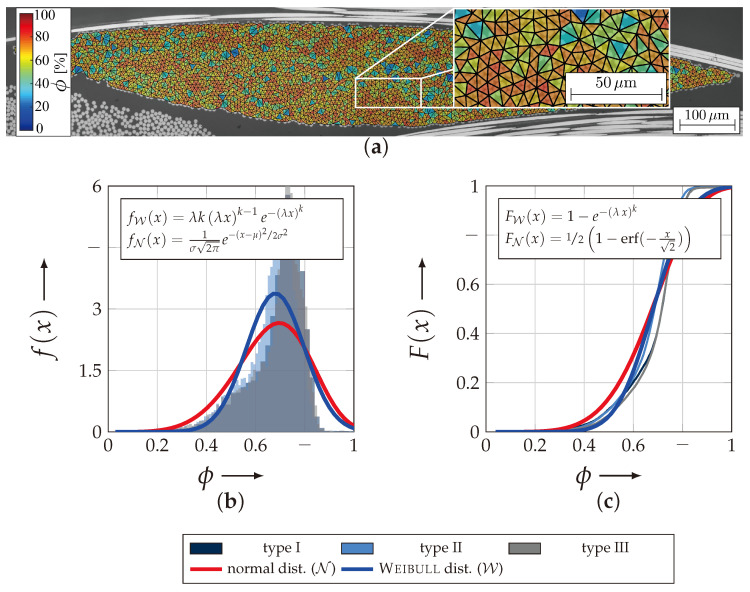
(**a**) Illustration of results of proposed algorithm for type III material; (**b**) histogram for each fabric type’s FVC (ϕ) and fit of different probability density functions (f(x)) over all data; (**c**) illustration of the cumulative probability functions (F(x)).

**Figure 12 materials-13-04772-f012:**
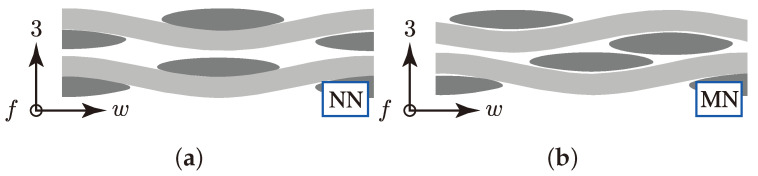
Illustration of the two idealistic configurations: (**a**) no nesting (NN) and (**b**) maximum nesting (MN) of textile reinforcements.

**Figure 13 materials-13-04772-f013:**
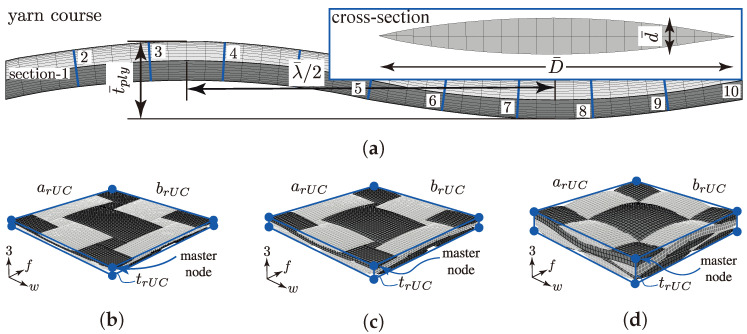
Illustration and notation of (**a**) geometry and mesh of the yarn course and cross-section as well as definition of sections (section-1, etc.) and rUC of (**b**) type I rUC${}_{I}^{\mathrm{MN}}$, (**c**) type II rUC${}_{\mathrm{II}}^{\mathrm{MN}}$ and (**d**) type III rUC${}_{\mathrm{III}}^{\mathrm{MN}}$ (maximum nesting configuration; dimensions are not representative—normalised to width).

**Figure 14 materials-13-04772-f014:**
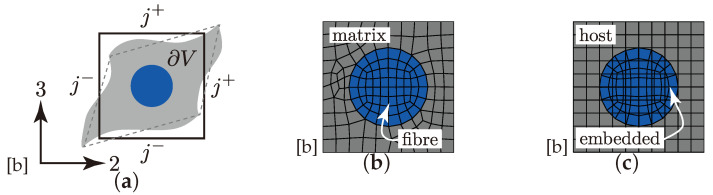
(**a**) Schematic illustration and notation of periodic boundary condition, (**b**) regular mesh and (**c**) embedded element mesh (exemplary for a fibre matrix rUC).

**Figure 15 materials-13-04772-f015:**
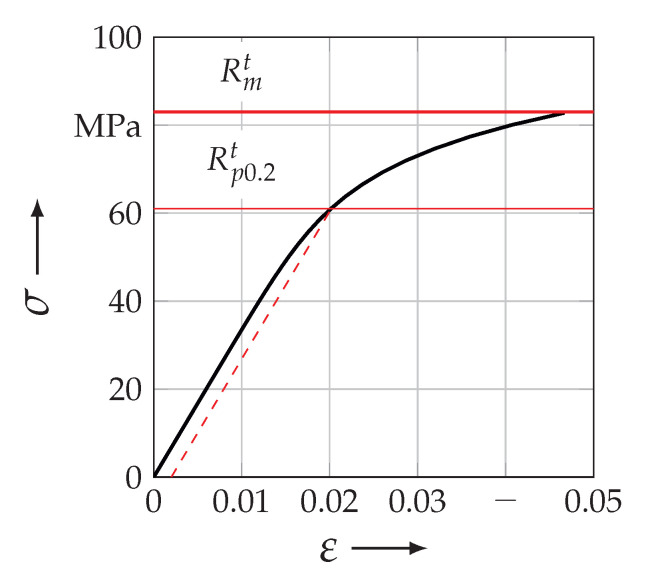
Underlying elastic–plastic stress–strain curve of RTM6-2 for numerical computation.

**Figure 16 materials-13-04772-f016:**
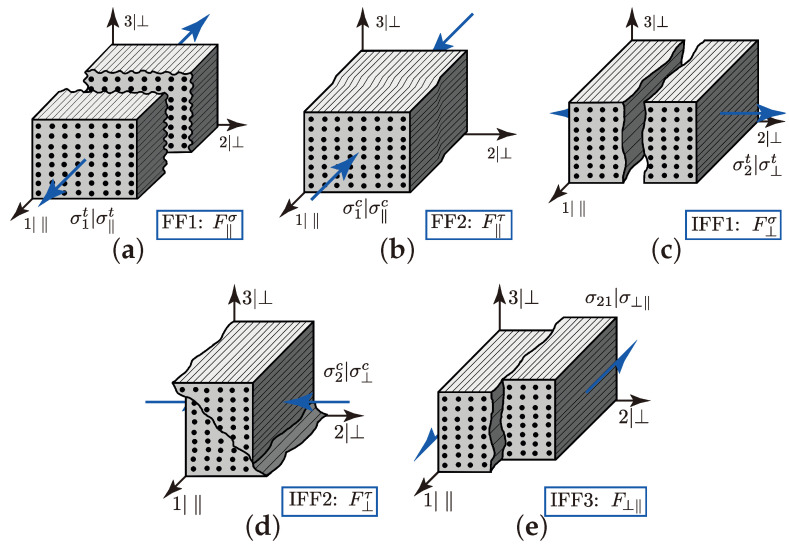
Illustration of failure modes (FMs) of a brittle transversely isotropic material: (**a**) fibre failure parallel tension (FF1: $F_{\|}^{\sigma }$), (**b**) fibre failure parallel compression (FF2: $F_{\|}^{\tau }$), (**c**) inter fibre failure transverse tension (IFF1: $F_{⊥}^{\sigma }$), (**d**) inter-fibre failure transverse compression (IFF2: $F_{⊥}^{\tau }$) and (**e**) inter fibre failure shear (IFF3: $F_{⊥\|}$) [38].

**Figure 17 materials-13-04772-f017:**
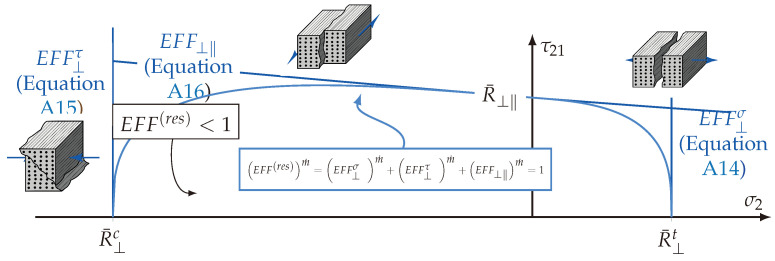
Schematic Illustration of Cuntze’s transverse fracture body for the 2D load vector $\left\{\sigma \right\}={\left(0,\sigma_{2},0,0,0,\tau_{21}\right)}^{⊤}$ and corresponding IFF Modes [39,71].

**Figure 18 materials-13-04772-f018:**
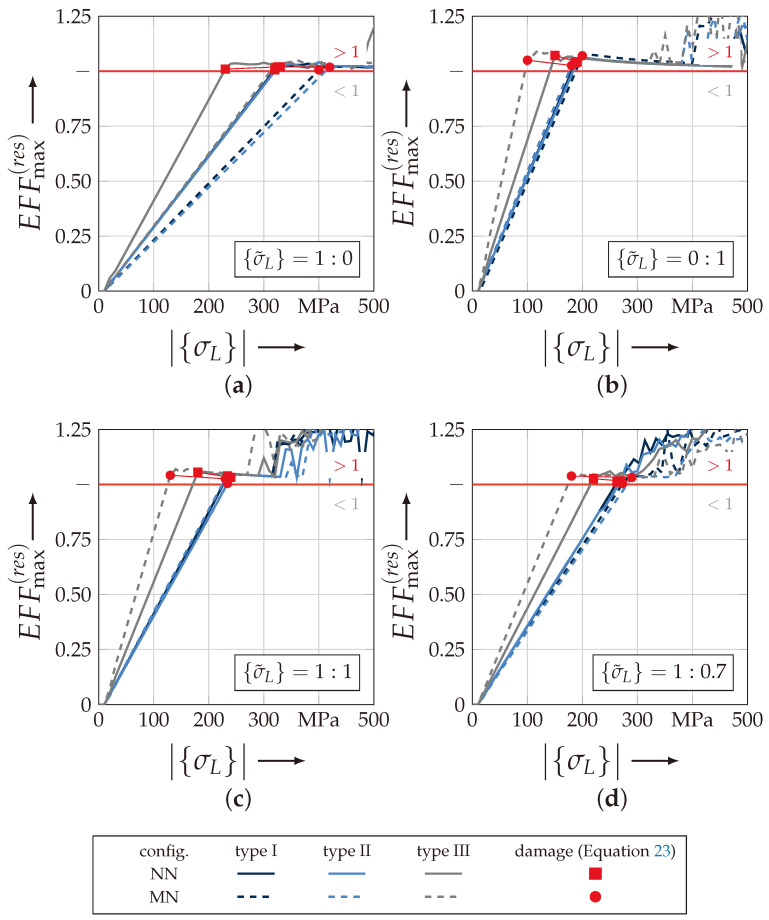
Results of the damage identification according to Equation (Equation 23) ($EFF_{\max}^{\left(\mathrm{res}\right)}$) of the considered rUCs at selected load paths $\{{\tilde{\sigma }}_{L}\}=$ (**a**) 1:0, (**b**) 0:1, (**c**) 1:1 and (**d**) 1:0.7.

**Figure 19 materials-13-04772-f019:**
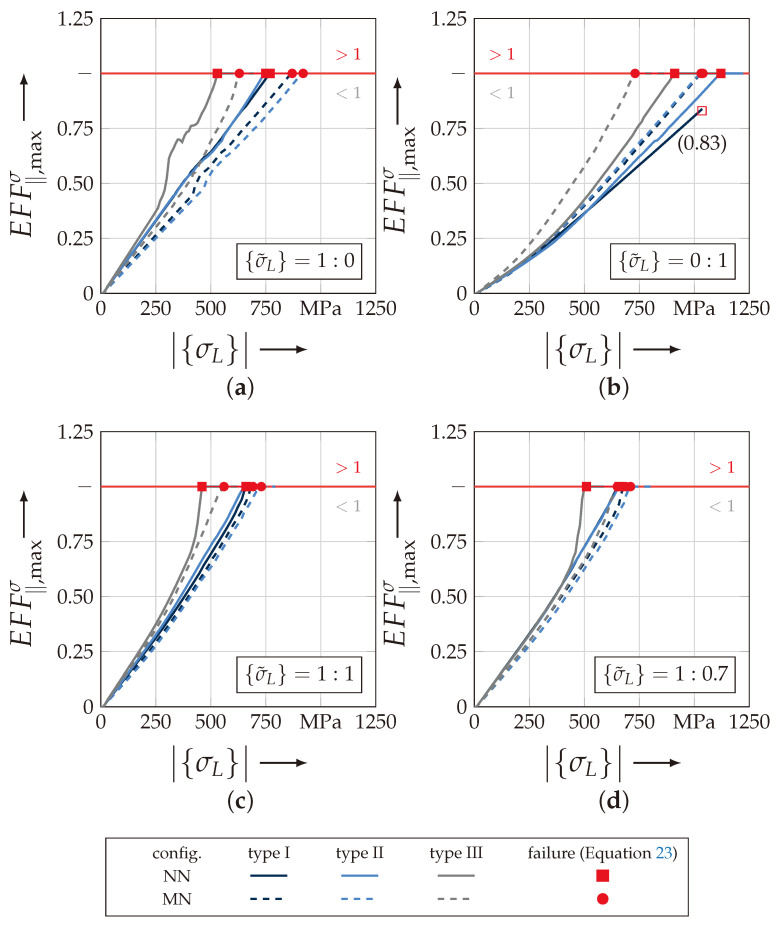
Results of the failure identification according to Equation (Equation 23) ($EFF_{\|,\max}^{\sigma }$) of the considered rUCs at selected load paths $\{{\tilde{\sigma }}_{L}\}=$ (**a**) 1:0, (**b**) 0:1, (**c**) 1:1 and (**d**) 1:0.7.

**Figure 20 materials-13-04772-f020:**
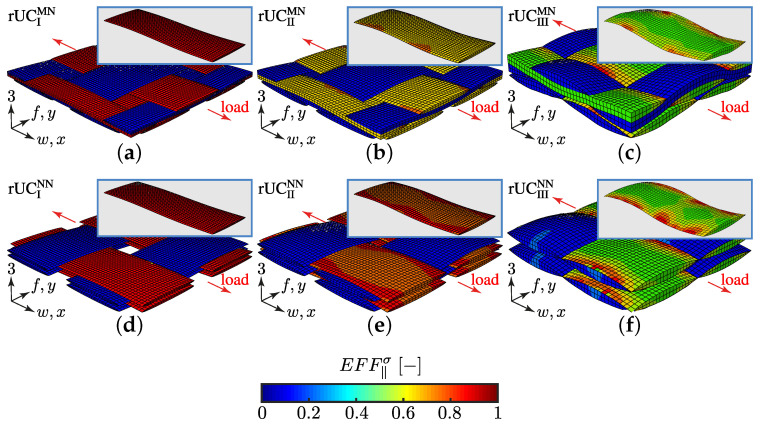
Exemplary illustration of the distribution of $EFF_{\|}^{\sigma }$ for the load path $\left\{{\tilde{\sigma }}_{L}\right\}$ = 1:0 for the considered rUC (**a**) rUC${}_{I}^{\mathrm{MN}}$, (**b**) rUC${}_{\mathrm{II}}^{\mathrm{MN}}$, (**c**) rUC${}_{\mathrm{III}}^{\mathrm{MN}}$, (**d**) rUC${}_{I}^{\mathrm{NN}}$, (**e**) rUC${}_{\mathrm{II}}^{\mathrm{NN}}$ and (**f**) rUC${}_{\mathrm{III}}^{\mathrm{NN}}$ and highlighting of a weft yarn.

**Figure 21 materials-13-04772-f021:**
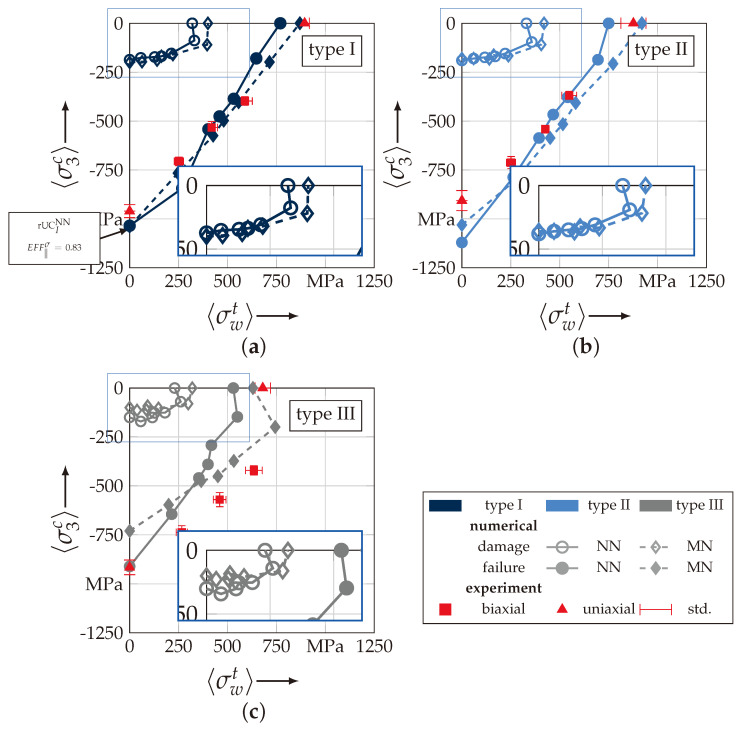
Comparison of experimentally and numerically determined fracture resistances for (**a**) type I, (**b**) type II and (**c**) type III composites.

**Figure 22 materials-13-04772-f022:**
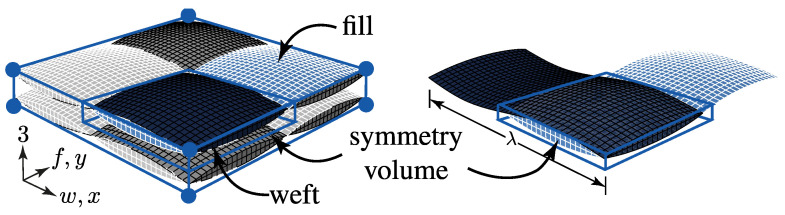
Illustration of assessed volume to identify the influence of the textile architecture.

**Figure 23 materials-13-04772-f023:**
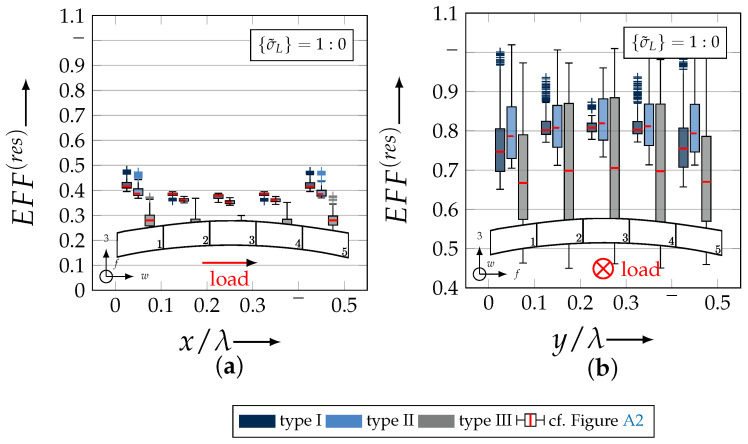
Illustration of resultant stress effort $EFF^{\left(res\right)}$ per defined normalised section according to Figure 13a at load ratio $\left\{{\tilde{\sigma }}_{L}\right\}$ = 1:0 for (**a**) weft and (**b**) fill yarn.

**Figure 24 materials-13-04772-f024:**
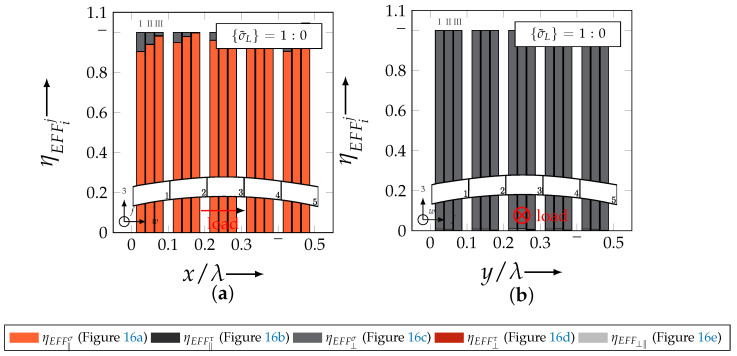
Illustration of the composition of FM by their fractions $\eta_{EFF_{i}^{j}}$ on the resultant stress effort for the load path $\left\{{\tilde{\sigma }}_{L}\right\}$ = 1:0 for (**a**) weft and (**b**) fill yarn.

**Figure 25 materials-13-04772-f025:**
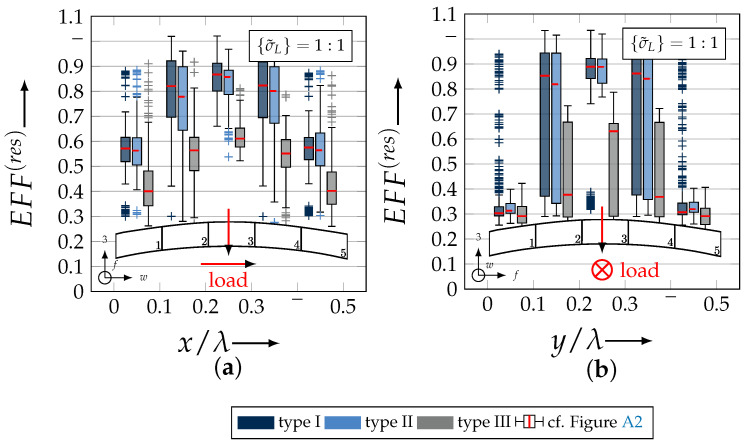
Illustration of resultant stress effort $EFF^{\left(res\right)}$ per defined normalised section according to Figure 13a at load ratio $\left\{{\tilde{\sigma }}_{L}\right\}$ = 1:1 for (**a**) weft and (**b**) fill yarn.

**Figure 26 materials-13-04772-f026:**
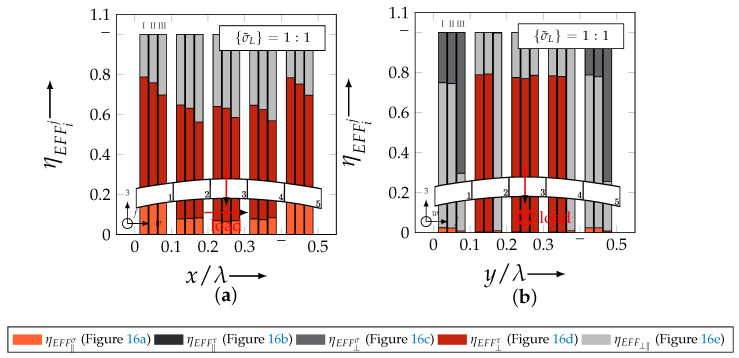
Illustration of the composition of FM by their fractions $\eta_{EFF_{i}^{j}}$ on the resultant stress effort for the load path $\left\{{\tilde{\sigma }}_{L}\right\}$ = 1:1 for (**a**) weft and (**b**) fill yarn.

**Figure 27 materials-13-04772-f027:**
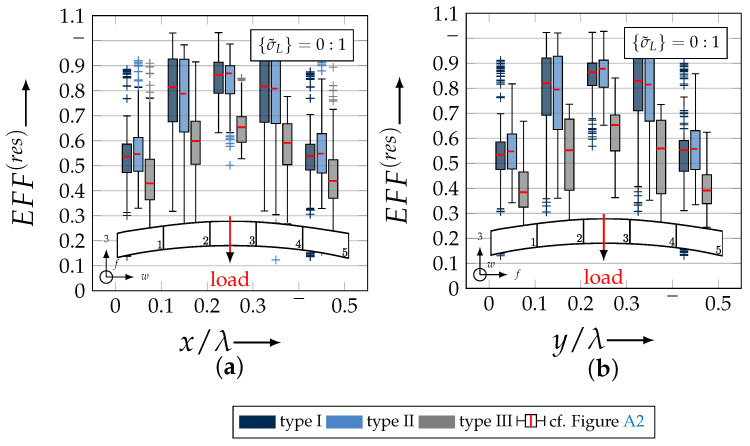
Illustration of resultant stress effort $EFF^{\left(res\right)}$ per defined normalised section according to Figure 13a at load ratio $\left\{{\tilde{\sigma }}_{L}\right\}$ = 0:1 (uniaxial out-of-plane compression loading) for (**a**) weft and (**b**) fill yarn.

**Figure 28 materials-13-04772-f028:**
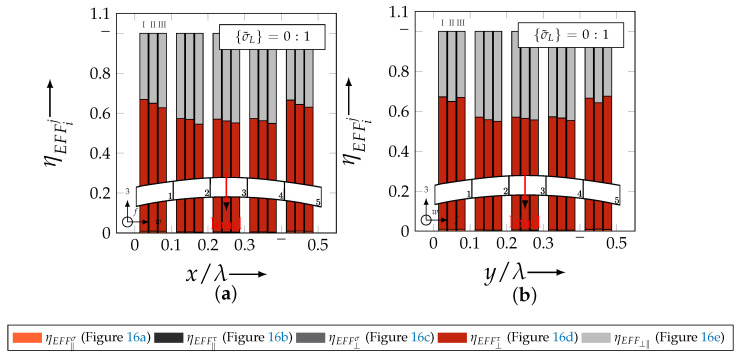
Illustration of the composition of FM by their fractions $\eta_{EFF_{i}^{j}}$ on the resultant stress effort for the load path $\left\{{\tilde{\sigma }}_{L}\right\}$ = 0:1 for (**a**) weft and (**b**) fill yarn.

**Figure 29 materials-13-04772-f029:**
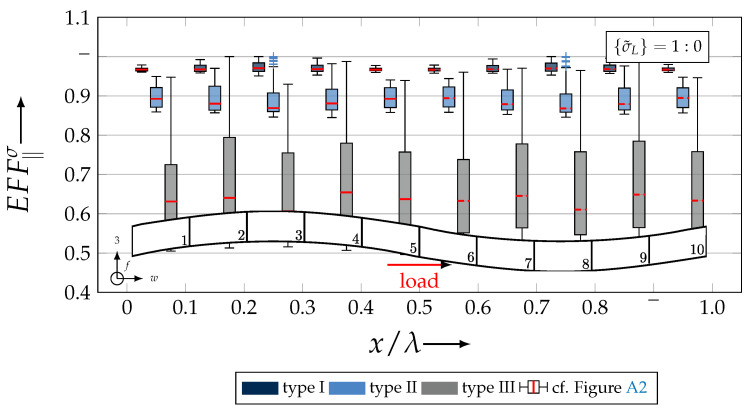
Illustration of stress effort $EFF_{\|}^{\sigma }$ per defined normalised section according to Figure 13a at load ratio $\left\{{\tilde{\sigma }}_{L}\right\}$ = 1:0 for weft yarn ($EFF_{\|}^{\sigma }$ did not occur in the fill yarn).

**Figure 30 materials-13-04772-f030:**
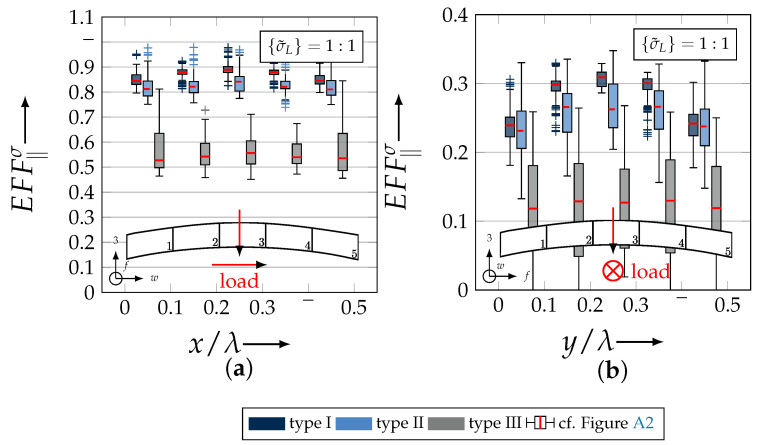
Illustration of stress effort $EFF_{\|}^{\sigma }$ per defined normalised section, according to Figure 13a, at load ratio $\left\{{\tilde{\sigma }}_{L}\right\}$ = 1:1 for (**a**) weft and (**b**) fill yarn.

**Figure 31 materials-13-04772-f031:**
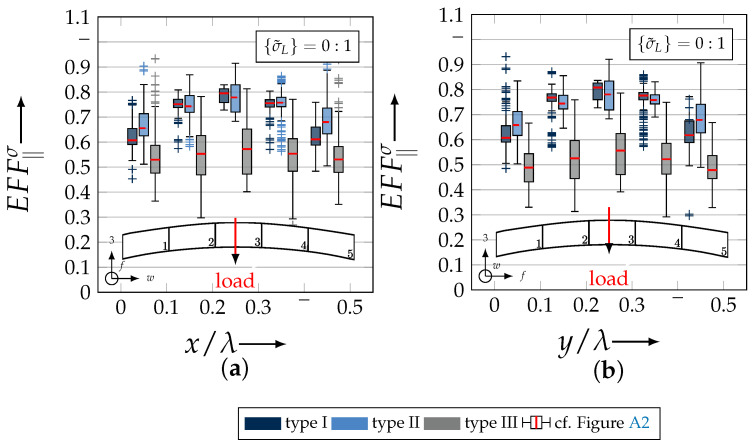
Illustration of stress effort $EFF_{\|}^{\sigma }$ per defined normalised section, according to Figure 13a at load ratio $\left\{{\tilde{\sigma }}_{L}\right\}$ = 0:1 for (**a**) weft and (**b**) fill yarn.

**Figure 32 materials-13-04772-f032:**
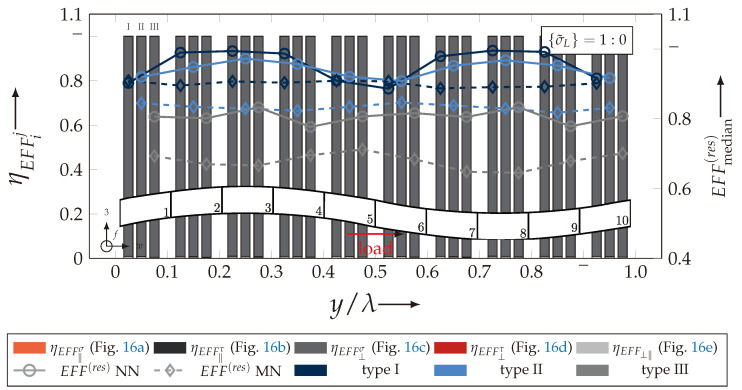
Illustration of the composition of FM by their fractions $\eta_{EFF_{i}^{j}}$ on the resultant stress effort of rUC${}_{I-\mathrm{III}}^{MN}$ and on median resultant stress effort $EFF_{\mathrm{median}}^{\left(res\right)}$ of MN and NN configs. at damage initiation for rUC${}_{I-\mathrm{III}}^{MN}$ at load path $\left\{{\tilde{\sigma }}_{L}\right\}$ = 1:0.

**Figure 33 materials-13-04772-f033:**
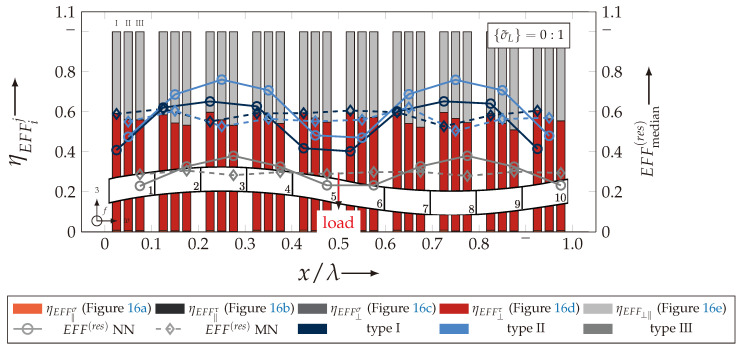
Illustration of the composition of FM by their fractions $\eta_{EFF_{i}^{j}}$ on the resultant stress effort of rUC${}_{I-\mathrm{III}}^{MN}$ and on median resultant stress effort $EFF_{\mathrm{median}}^{\left(res\right)}$ of MN and NN configs. at damage initiation for rUC${}_{I-\mathrm{III}}^{MN}$ at load path $\left\{{\tilde{\sigma }}_{L}\right\}$ = 0:1.

**Figure 34 materials-13-04772-f034:**
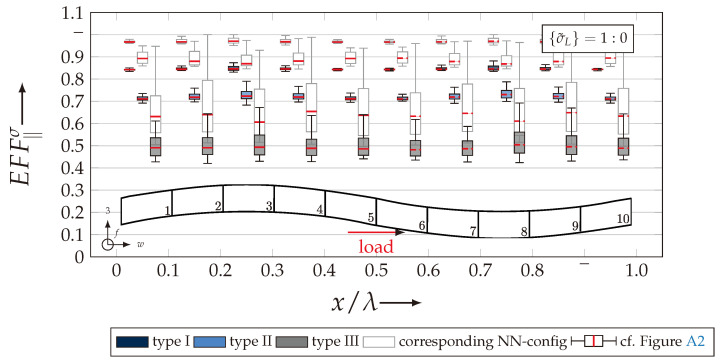
Comparison of $EFF_{\|}^{\sigma }$-distribution for NN and MN configs. for the triggered fracture resistance of the NN config. according to Equation (Equation 23) at load ratio $\left\{{\tilde{\sigma }}_{L}\right\}$ = 1:0 in weft direction for considered weft yarn according to Figure 22.

**Figure 35 materials-13-04772-f035:**
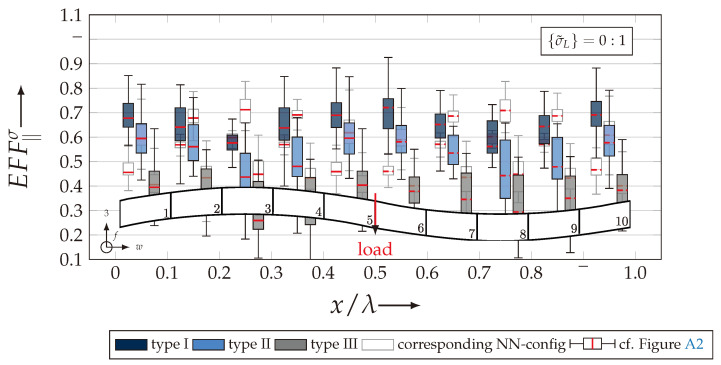
Comparison of $EFF_{\|}^{\sigma }$-distribution for NN and MN configs. for the triggered fracture resistance of the MN config., according to Equation (Equation 23) at load ratio $\left\{{\tilde{\sigma }}_{L}\right\}$ = 0:1 in weft direction.

**Table 1 materials-13-04772-t001:** Selected properties of ECC-style fabric reinforcements with Tenax^®^ HTA 40 3K (200 tex) yarns.

Type	ECC^™^-Style [−]	Weight $\left[g\;m^{-2}\right]$	Setting [Threads/cm]
I	447	160	4
II	450	200	5
III	470	285	7

**Table 2 materials-13-04772-t002:** Selected properties of a Tenax^®^ HTA filament [50].

$d_{f}$ [µm]	$\varrho_{f}$ $\left[g\;m^{-3}\right]$	$E_{f\|}$ [GPa]	$E_{f⊥}$ [GPa]	$G_{f\|⊥}$ [GPa]	$\nu_{f⊥\|}$ [−]	$R_{f\|}$ [MPa]	$\varepsilon_{f\|,\max}$ [−]
7.00 ${}^{†,‡}$	1.77 ${}^{‡}$	240 ${}^{‡}$	28 ${}^{†}$	50 ${}^{†}$	0.230 ${}^{†}$	4100 ${}^{‡}$	1.700 × 10^−2^ ${}^{‡}$

${}^{†}$ [50], ${}^{‡}$ [51].

**Table 3 materials-13-04772-t003:** Exemplary results of normal distribution fits for the crimp angel $\beta_{c}$ and the nesting factor $\eta_{P}$ (with $\mu_{N}$: mean value and $\sigma_{N}^{2}$: variance).

Type	$\beta_{c}$			$\eta_{P}$		
	$\mu_{N}$ [°]	$\sigma_{N}^{2}$ [°]	ν [%]	$\mu_{N}$ [−]	$\sigma_{N}^{2}$ [−]	ν [%]
I	4.385	1.765	30.297	0.831	2.84 × 10^−3^	6.413
II	5.916	1.121	17.897	0.842	1.362 × 10^−3^	4.383
III	12.134	2.889	14.008	0.868	1.364 × 10^−3^	4.255

**Table 4 materials-13-04772-t004:** Expectancy value (μ) and variance ($\sigma^{2}$) of Weibull (W)- and normal (N)-distribution approximation of the random variable FVC ($\phi^{m}$).

Data	$\mu_{N}$ [−]	$\sigma_{N}^{2}$ [−]	ν [%]	$\mu_{W}$ [−]	$\sigma_{W}^{2}$ [−]	ν [%]
I	0.692	0.014	17.098	0.679	0.023	22.369
II	0.665	0.013	17.145	0.653	0.020	21.660
III	0.684	0.015	17.906	0.672	0.023	22.607
I + II + III	0.681	0.014	17.375	0.726	0.022	20.522

**Table 5 materials-13-04772-t005:** Geometry parameters of yarn course and cross section and consequentially of the rUCs in NN and MN configuration of each type.

Type	Config.	Course		Cross Section		rUC		
		$\bar{\lambda }$ [µm]	${\bar{t}}_{\mathrm{ply}}$ [µm]	$\bar{D}$ [µm]	$\bar{d}$ [µm]	$a_{\mathrm{rUC}}^{†}$ [mm]	$t_{\mathrm{rUC}}$ [µm]	$\eta_{P}$ ${}^{‡}$ [−]
I	NN	5.216 × 10^3^	1.500 × 10^2^	1.881 × 10^3^	0.750 × 10^2^	5.216	3.000 × 10^2^	(1) 0.852
	MN						2.555 × 10^2^	
II	NN	3.746 × 10^3^	1.900 × 10^2^	1.577 × 10^3^	0.950 × 10^2^	3.746	3.800 × 10^2^	(1) 0.874
	MN						3.321 × 10^2^	
III	NN	2.548 × 10^3^	2.700 × 10^2^	1.254 × 10^3^	1.350 × 10^2^	2.548	5.400 × 10^2^	(1) 0.870
	MN						4.697 × 10^2^	

${}^{†}\;\equiv b_{rUC}$, ${}^{‡}\;$ determined by $t_{rUC}$ to $t_{rUC}^{\;\mathrm{NN}}$ according to Equation (Equation 3).

**Table 6 materials-13-04772-t006:** Elastic properties and strength of RTM6-2 from Hexcel^®^[46].

$E_{m}$ [MPa]	$\nu_{m}$ [−]	$G_{m}^{†}$ [MPa]	$R_{p0.2}$ [MPa]	$R_{m}$ [MPa]
3354	0.380	1215	61	82

${}^{†}$$G=\frac{E_{m}}{2(1+\nu_{m})}$.

**Table 7 materials-13-04772-t007:** Instaneous linear-elastic properties for a transverse isotropic CRFP with an average FVC $\bar{\phi }$ of 72% (micromechanical semi-analytical solution; Section B.1).

	$E_{\|}$ [GPa]	$E_{⊥}$ [GPa]	$G_{\|⊥}$ [GPa]	$G_{⊥⊥}$ [GPa]	$\nu_{⊥\|}$ [−]	$\nu_{⊥⊥}$ [−]
value	172.299	18.548	9.280	6.941	0.272	0.336
Equation	(Equation 31)	(A6)	(A7)	(A8)	(A9)	(A10)

**Table 8 materials-13-04772-t008:** Strengths and Cuntze’s parameters for a transversal isotropic CRFP with an average FVC of ${\bar{\phi }}^{\left(m\right)}$ = 72%.

${\bar{R}}_{\|}^{t}$ [MPa]	${\bar{R}}_{\|}^{c}$ [MPa]	${\bar{R}}_{⊥}^{t}$ [MPa]	${\bar{R}}_{⊥}^{c}$ [MPa]	${\bar{R}}_{⊥\|}^{}$ [MPa]	$\dot{m}$ [−]	$b_{⊥\|}$ [−]	$b_{⊥}^{t}$ [−]
2602	1152	70	211	98	2.600	0.260	1.050

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
