# Peer review of "Determining the Damage and Failure Behaviour of Textile Reinforced Composites under Combined In-Plane and Out-of-Plane Loading"

_materials, 2020, doi:10.3390/ma13214772_

Round 1

Reviewer 1 Report

  1. “Figure 1. Elaborated (swt /s3c)-testing for textile-reinforced composites:

      a) schematic test set-up and global and material CS;

      b) technical drawing with characteristic dimensions in [mm],”

  1. Do the dimensions of the test specimens satisfy the standard?

  1. “Figure 3. a) Illustration and definition of evaluation areas of the 2D strain field measurement with DIC” Please add scale bars.
  1. Conclusion

       Please list the major findings using bullet points.

Author Response

Dear reviewer.

Thank you very much for your comments and suggestions. 

Please find my responses to your points following up:

1.

I am not sure what you want to point out with this comment or suggestion?

In case you suggested a better highlighting of the sub captions, I incorporated a), b), etc. in bold font to avoid too much line breaks in the whole paper. Could that satisfy you?

2. 

Yes, the dimensions for the uniaxial test method for in-plane tension satisfy the used standard. Please refer to DIN EN 527-4 as cited in the paper. This standard resembles the ASTM-standard D3039. Following dimensions are recommended width 25–50 mm, thickness 2-10 mm, length >250mm and gauge lengths 150 mm. I chose width 25mm, thickness 4mm and the length according to standard. I added these dimensions in the paper by using a footnote.

Testing standards for the out-of-plane compression and the biaxial testing are currently not available. These testing methods are newly developed as described in the paper.

3. 

Scale bars were added also for Fig. 4 and 6. Thank you for the advice.

4.

major findings are already listed in bullet points. We made some minor changes

Reviewer 2 Report

Dear Editor, 

the manuscript is very good written, even if there are some minor issues that can be improved. The major issue, in my opinion, is page range. This publication should be published as a book, not as a regular article.

Author Response

Dear reviewer.

Thank you very much for your comments and suggestions. Yes we thought about to split it in a numerical and experimental part. 

Maybe the editor gives us the opportunity.

Reviewer 3 Report

This manuscript presents a very interesting work, dealing with the development of experimental and numerical methods for identification of the influence of textile reinforcement when subjected to combined in-plane and out-of-plane loads. The authors complemented this experimental investigation with a numerical simulation. The topic is within the scope of the journal and this study is important and addresses an important issue. However further explanations are needed. The authors are also advised to check the manuscript again carefully in order to avoid English grammar errors.

I recommend major revision of this manuscript. Observations and clarifications are required before the manuscript is suitable for publication, as outlined in the following points.

Abstract:

  • Lines 10-14: Please revise sentence. Sentence is too long.

Section 1

  • The literature review is too short. Some experimental studies were recently carried out concerning the application of TRM strengthening solutions using CFRP textile meshes in full-scale specimens. First the authors have to distinguish if are talking about material characterization or real performance of the retrofit application using this technique.
  • Additionally, the authors should include in the motivation of this study the development of efficient retrofitting to prevent the out-of-plane failure of masonry infill walls. Some experimental studies should be slightly mentioned, highlighting the different variables that justify the high seismic vulnerability of these elements (10.1080/13632469.2018.1453400; 10.1016/j.conbuildmat.2018.10.011)
  • Lines 80-85 Please revise sentence

Section 2

  • Did the authors observed any stress accumulation in the region near to the attachment of the specimen? Pictures of the real test setup are welcome. More details are needed concerning this methodology. Which aspects are new relative to other existing methodologies?
  • Table 2: Is there any coefficient of variation available? Did the authors tested this material or these properties are provided by the supplier?
  • Table 3: A graph containing these results would be interesting in order comparing the properties of the three materials.
  • Figure 5: Please consider changing the colors for a more evident interpretation of the results.
  • More details should be included concerning the failure modes observed in these tests. The discussions needs to be improved.
  • Figure 7: The plots looks fine, however again the similar colors are not friendly for a correct interpretation. Other types of distributions were tested?
  • Table 5: It is not explicit the average calculation. The coefficient of variation needs to be included and discussed.

Section 3

  • The authors are invited to discuss about the mesh adopted. Did the mesh affect significantly the numerical results? If yes, please detail.
  • Line 412: What does the authors mean with “sufficient accordance”? Did the authors consider any other variation? If yes, justify it.
  • Figure 20: The numerical results presents slight distance from the experimental results. Why? The quantification of this difference are welcome.

Conclusions

  • Please include additional future works

Author Response

Dear Reviewer.

Thank you for your comments and suggestions.

Please find my reply for each specific point following up:

Abstract:

In Lines 10-14 there are 2 sentences. I splitted the sentences into several shorter ones. Made some minor changes

Section 1

Dear reviewer, I am concerned that your mentioned topics are not fully compliant with the addressed topics in the paper. As I wrote and restricted the field of topic: “Hereby reducing the structural weight of trains, cars, airplanes, etc. is one of the key factors in reaching that goal. Using optimised carbon fibre reinforced plastics (CFRP) can lead to a significant weight reduction, which can consequently decrease carbon emissions up to 20%.” the construction industry and retrofitting of masonry, respectively, is not emphasised in this paper. Nevertheless, because it is an interesting field and colleagues of mine are also working on that topic, I incorporated a sentence about the general usage of CFRP. So you can read: “The applications and research projects of CFRP cover  the construction industry with e.g. replacing steel reinforcements by lignin-based carbon fibre reinforced thermosets~\cite{boehm_cs3_2018} or the retrofitting of masonry structures with carbon fibre meshes~\cite{Kariou_trm_2018,ricci_masonry_2018}, the transportation industry with e.g. hybrid fibre reinforced thermoplastic hollow drive-shafts~\cite{wuerfel_tube_2020}, up to high-performance applications in the aviation and space industry”. In this sentence, I incorporated one of the suggested papers (10.1016/j.conbuildmat.2018.10.011). Additionally, I added the paper of Kariou et.al. (10.1016/j.compositesb.2019.04.026). Very interesting work of TRM strengthening solutions. In my opinion this paper is more compliant to my paper than the suggested (10.1080/13632469.2018.1453400), because in that there is no evidence of any fibre reinforced composite.

Section 2

  • Hopefully you mean by “attachement” the compression stamps in the paper. As described in Section 3.1: „…and due to the choice of large radii it reduces the effect of stress peaks in the vicinity of the radii due to Hertzian contact“. We were aware of the effects of the stress concentrations. The numerical and analytical calculations about the stress concentration have been presented in [9] and in [22]. In case you are still concerned about stress concentration in the vicinity of the stamps, compare the strain distribution of Fig. 4b. There is no evidence of a accumulation of higher strains near the “attachment”.

    The “real test set-up” is illustrated in Fig. 3, would that suit you? Fig. 4 illustrates the testing set-up in the gauge section. The aspects that are new to this methodology is explained in the introduction: “Most other testing methods for biaxial load application are restricted to prescribed load paths by geometry, have demanding requirements on the specimen design and preparation or cannot depict the size and the influence of the textile specific ondulation [23,24]”

  • These are the properties tested by the supplier (Teijin) and published in Schürmann page 42. I added the reference in Table 2 accordingly. Coefficients of variation are not available in these references.

  • Yes, you’re right that a graphical interpretation of the results is easier to compare. But the emphasis of the paper is not on the stiffness properties rather more on the fracture behavior. To avoid unnecessary elongation of the paper I inserted a bar plot and moved the table in the appendix and referenced it in the paper section.

  • Added some sentences concerning the failure modes

  • No, other distributions were not tested on the data of Figure 7. The xi-square test on a Normaldistribution for every data type neglected the Nullhypothesis, consequently it can be assumed that the distribution is “Normal”.

  • As written in the section the results concern the expectancy value not the arithmetic mean value. Calculated accordingly to the chosen distributions (Normal and Weibull). In case of the Normaldistribution the mean and expectancy value are identical because of the symmetric distribution. For any Weibull-distribution (k unequal 3.602; Weibull converges to a Normal-distribution) the average and the expectancy value are not equal. In Table 4 (former Table 5) the variance (sigma^2) of each distribution, calculated accordingly, are given. As the CoV is calculated as the ratio of the standard deviation to the mean value, I added the CoV but didn’t include it in the discussion.

Section 3

  • No, it didn’t significantly. Just because of the feature of a structured mesh and high mesh quality we chose the embedded element method. I wrote:” All parts have been meshed using ABAQUS and C3D8 and C3D8R elements for the embedded and the host parts, respectively.“ I added further: “The mesh size was set identical to \SI{0.1}{\mm} for all rUCs. Any significant influence of the mesh size on the obtained results was not found” to emphasize it.

  • The usage of “Sufficient” may be a miss translation from German. I changed it to “good”.  What do you mean by any other variation in this Section. This section explains the failure mode concept by Cuntze and the table lists the values used for the numerical implementation. The values are in good accordance to the results in [70]. Nevertheless, the variation of the FVC is seen more by type III and could affect the results more than presumed. This effect should be thoroughly incorporated in the next work.

  • Added the relative deviation from the mean values of experiment to each determined failure resistance of the rUCs in Nesting and NoNesting configuration. To avoid incomprehensibility  I only added the values in the Table in the appendix. I incorporated some important values in the text

Conclusion

added some ideas of future works

Reviewer 4 Report

GENERAL

The engineering teams specialised in the design of structural pieces made of polymer composites reinforced with carbon fibre textile fabrics will find quite interesting this paper, especially if the pieces are subjected to complex internal stress states and need to have low sensibility for initial or limited damage. The authors present an extensive, rigorous report on the in-depth behaviour of specimens fabricated with three types of carbon fibre fabrics in polymer matrices analysed under simultaneous loading in two perpendicular directions by the use of a biaxial loading testing device that allows different combinations of loads in either directions. The main interest of the authors is to assess the influence of the fabric architecture on ondulation and on the crimp, nesting effects and interlocking of adjacent fabric layers. The composite configuration and the stress conditions are analysed in three level scales (macro, meso and micro) using statistical analysis and numerical algorithms for damage and failure behaviour and assessing global homogeneous stiffness and strength values. An extension to 3D failure modes for brittle transversally isotropic material as supposed for the studied samples is also proposed for damage identification under selected load paths. In short is a complete laboratory and theoretical study on the matter. 

The text organization is very good with appendices of formulations, testing results and abbreviations. Language is quite good, although is sometimes difficult to follow due to complex notation of some variables, the presence of different techniques and models and of the inherent complexity of some discussions. Also some minor formal details are at least surprising to the reader as the ones listed below.

DETAIL COMMENTS

  1. The use of the word “respectively” sounds weird to me in many instances in the text like in lines 5, 31, 59, 79, 123, 148, 162, 164, 167, 368, and 728 among others. Also the abbreviation “cf.” for addressing figures and tables is not of common use.
  2. Line 42: could?
  3. Line 58: an advanced methods?
  4. Line 71: improve wording
  5. Line 91: Please place figure 10 at the beginning of the text for better understanding of the axis systems and notation.
  6. Line 130: could been?
  7. Line 148: listed in 7?
  8. Line 168: Although the parameters of Table 3 are cited in the text, a definition list would be useful.
  9. Line 254: The closest two-digit value is 91% not 93%.

Author Response

Dear reviewer.

Thank you very much for your comments and suggestions.

  1. Yes, we changed it
  2. verb changed to “have” and auxiliary verb changed to “been”. Further grammar and spelling checks included in the paper.

  3. changed “by an advanced methods” to “by methods and algorithms”

  4. Yes, I changed it
  5. Thanks for the advice. Figure 10 was placed prior and was extended with the experimental multiscale approach. This should improve the understanding of the two approaches. It is a major change in the paper

  6. grammar: changed to was investigated

  7. added Tab.

  8. The Table has been moved in the Appendix. A Figure was placed instead. We Found the results are better comparable in a Figure.
  9. Yes, you’re right. Changed it

Round 2

Reviewer 3 Report

The authors considered all the revisions and suggestions made by the reviewers. The manuscript quality increased and is now suitable for publication.